# HALO: Human-Aligned End-to-end Image Retargeting with Layered Transformations

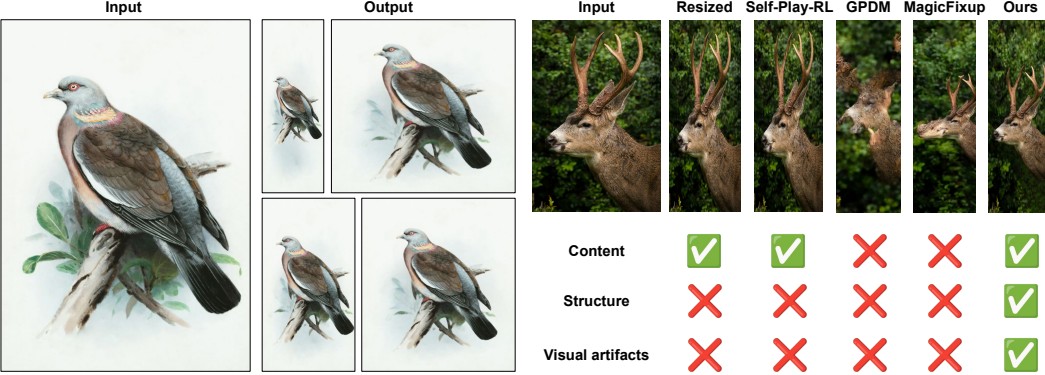

Figure 1: **Content- and structure-aware image retargeting.** Our method, HALO, takes a single image as input and reformats it for different aspect-ratios. Compared to previous methods (Kajiura et al., 2020; Elnekave & Weiss, 2022; Alzayer et al., 2024), our method shows better performance in preserving the structure and content of the input image and has less visual artifacts.

## Abstract

Image retargeting aims to change the aspect-ratio of an image while maintaining its content and structure with less visual artifacts. Existing methods still generate many artifacts or lose a lot of original content or structure. To address this, we introduce HALO, an end-to-end trainable solution for image retargeting. The core idea of HALO is to warp the input image to target resolution. Since humans are more sensitive to distortions in salient areas than non-salient areas of an image, HALO decomposes the input image into salient/non-salient layers and applies different wrapping fields to different layers. To further minimize the structure distortion in the output images, we propose perceptual structure similarity loss which measures the structure similarity between input and output images and aligns with human perception. Both quantitative results and a user study on the RetargetMe dataset show that our algorithm achieves SOTA. Especially, our method increases human preference by $13.21\%$ compared with the second best method.

## 1 Introduction

Images are displayed on a diverse set of platforms and devices, each with a different aspect-ratio. Content creators are often required to produce multiple versions of the same image in different aspect-ratios, a task that becomes increasingly burdensome with the growing number of platforms. Resizing or cropping images are traditional approaches for it, but resizing can distort structures, and cropping inevitably removes content. Image retargeting (Rubinstein et al., 2010; Tang et al., 2019) seeks to address these problems and adjusts an image's aspect-ratio while preserving its key content and structure. As defined by (Rubinstein et al., 2010; Vaquero et al., 2010), a successful image retargeting outcome is as follows: (a) The key *content* in the input image should be preserved in the output image; (b) The inner *structure* of the input should be maintained in the output; (c) There should be *no distortion* or *visual artifacts* in the output image.

Many image retargeting algorithms have been proposed, including traditional optimization approaches (Liu & Gleicher, 2005; Setlur et al., 2005; Wolf et al., 2007; Simakov et al., 2008; Rubinstein et al., 2009; Barnes et al., 2009; Pritch et al., 2009; Rubinstein et al., 2010; Chen et al., 2010; Shi et al., 2010), weak- or self-supervised learning (Cho et al., 2017b; Tan et al., 2019), reinforcement learning (Kajiura et al., 2020), and generative modeling methods (Elnekave & Weiss, 2022; Granot et al., 2022). However, these methods still struggle to preserve both content and structure or generate less visual artifacts (*e.g.,* out-of-boundary, or OOB, artifact) as shown in Figure 2.

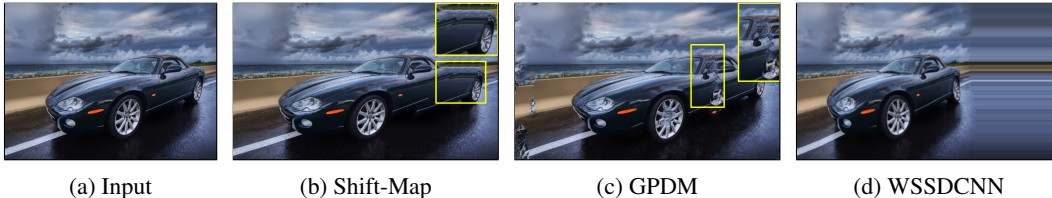

| (a) Input | (b) Shift-Map | (c) GPDM | (d) WSSDCNN |

Figure 2: **Limitations of exisiting retargeting methods.** Previous image retargeting methods have difficulty preserving the input image content and structure. (b) A traditional method Shift-Map (Pritch et al., 2009) duplicates the structure of the car. (c) A generative modeling method GPDM (Elnekave & Weiss, 2022) adds extra content. (d) A feed-forward method WSSDCNN (Cho et al., 2017b) introduces out-of-boundary (OOB) artifacts.

To address these problems, we propose HALO (Human-Aligned Layered transfOrmations for image retargeting), an end-to-end trainable model. The key idea behind HALO is to warp the input image to a target resolution with layered transformations. Recognizing that humans are more sensitive to distortions in salient regions than in non-salient areas, HALO decomposes the image into salient and non-salient layers based on a saliency map and applies different transformations to each layer. This design enables HALO to preserve critical details in salient regions while handling non-salient areas, and it also avoids OOB issues.

To further reduce the structure and content loss in output images, we use perceptual loss function as weak supervision to guide the algorithm to produce images close to the original image's content and structure. DreamSim (Fu et al., 2023), which emphasizes mid-level structure and distortion, is well-suited as a perceptual loss for image retargeting. However, since DreamSim is trained on square images, it cannot be directly applied to image retargeting. To address this, we develop a *layout augmentation* technique that adapts DreamSim for image retargeting and we introduce a new loss function, Perceptual Structure Similarity Loss (PSSL), which aligns closely with human perception.

Our contributions are as follows:

- A novel end-to-end trainable image retargeting algorithm based on layered transformations.
- A new Perceptual Structure Similarity Loss function for image retargeting tasks, aligning well with human perception.
- Extensive quantitative results and a user study on the RetargetMe dataset demonstrate that HALO achieves SOTA performance, with HALO significantly outperforming the second-best approach in the user study.

## 2 RELATED WORK

**Image Retargeting.** Image retargeting is a task to generate images with arbitrary aspect-ratios given an input image. Over the years, various approaches have been proposed, including conventional optimization-based methods (Rubinstein et al., 2008; 2009; Barnes et al., 2009; Simakov et al., 2008; Wolf et al., 2007; Pritch et al., 2009; Wang et al., 2008; Karni et al., 2009), weakly-supervised learning (Cho et al., 2017a; Tan et al., 2019), deep reinforcement learning (Kajiura et al., 2020), GAN based models (Shaham et al., 2019; Shocher et al., 2019; Hinz et al., 2021; Zhang et al., 2022), Patch Nearest Neighbor (PNN) (Granot et al., 2022; Elnekave & Weiss, 2022), and diffusion models (Wang et al., 2022; Kulikov et al., 2023; Zhang et al., 2023; Nikankin et al., 2023). Compared to optimization-based methods, we train an end-to-end model and it has *faster* inference

speed. Compared to end-to-end methods, our method uses layered transformations and predicts multiple warping flows, avoiding out-of-boundary issues and preserving salient contents better.

**Layered representations.** Layered representations (Lu et al., 2020; 2021; Yang et al., 2021) enable more flexible manipulation for an image or a video on different layers. It has been widely used for both images (He et al., 2009; Gandelsman et al., 2019) and videos (Lu et al., 2020; Liu et al., 2021; Lu et al., 2021; Kasten et al., 2021; Lee et al., 2023). We adopt the idea of layered representations and use it in the image retargeting task. It avoids out-of-boundary issues in the previous methods.

**Perceptual losses.** With the revolution of deep-learning, many pretrained networks (Krizhevsky et al., 2012; Simonyan & Zisserman, 2014; He et al., 2016a), can extract meaningful features from the images. Defined by measuring the feature distances, learning-based metrics (Dosovitskiy & Brox, 2016; Johnson et al., 2016; Zhang et al., 2018; Prashnani et al., 2018) show better alignment with human perception than the classic ones. More recently, DreamSim (Fu et al., 2023) is proposed to capture the mid-level similarities, such as structure and layout, between images. Perceptual losses are also used in image retargeting (Cho et al., 2017b; Tan et al., 2019) in the absence of paired training data. We use DreamSim, a perceptual loss focusing on the mid-level features such as structures and layouts. We find previous perceptual loss functions (*e.g.*, LPIPS (Zhang et al., 2018)) have difficulties handling structure distortions. We further adapt DreamSim to the image retargeting task by proposing a layout augmentation.

## 3 METHODOLOGY

### 3.1 OVERVIEW OF HALO

Figure 3 shows the framework of our method. HALO takes an image $I \in \mathbb{R}^{H \times W}$ and its saliency map $M$ as inputs to predict an output image $I' \in \mathbb{R}^{H' \times W'}$, where $H, W$ are the input height and width, $H', W'$ are the output height and width. The saliency map is a heatmap that measures the importance of pixels in the input image. The saliency map could be generated by a saliency detector (Gao et al., 2024), a segmentation network (*e.g.,* SAM (Kirillov et al., 2023)), or a user-defined mask. In this paper, we use saliency maps predicted by an off-the-shelf salient detector, MDSAM (Gao et al., 2024). Given a saliency map $M$, we decompose the input $I$ into a salient layer as $I_{SL} = I \odot M$ and a non-salient layer $I_{NSL} = I \odot (1 - M)$, where $\odot$ is the element-wise multiplication. To fill in the holes of the non-salient layer, we inpaint it with an off-the-shelf inpainting model (Suvorov et al., 2022): $I_{NSLI} = \text{Inpaint}(I_{NSL})$.

The reason for decomposing an image into two layers is based on the observation that a single transformation, as in (Cho et al., 2017b; Tan et al., 2019), cannot handle both salient and non-salient contents simultaneously well and may result in out-of-boundary (OOB) issues as shown in Figure 2 and Figure 7. A single transformation is able to preserve the salient content to the new target size, but may warp the non-salient pixels in an undesired way. Applying multiple transformations gives the model more flexibility to achieve retargeting without suffering from the OOB issues (Figure 7). We finally formulate the output image $I'$ as

$$I' = \text{Warp}(I_{SL}, \mathcal{F}_{SL}) \odot M' + \text{Warp}(I_{NSLI}, \mathcal{F}_{NSL}) \odot (1 - M'), \tag{1}$$

where $\mathcal{F}_{SL}, \mathcal{F}_{NSL} \in \mathbb{R}^{H' \times W' \times 2}$ are vector warping fields predicted by our Multi-Flow Network (MFN), and the warped saliency map $M' = \text{Warp}(M, \mathcal{F}_{SL})$.

### 3.2 MULTI-FLOW NETWORK

Inspired by Spatial Transformer Networks (STNs) (Jaderberg et al., 2015; Peebles et al., 2022; Ofri-Amar et al., 2023), we design a Multi-Flow Network (MFN) shown in Figure 3. Our MFN consists of an encoder, $L$ cross-attention blocks, and two heads to predict the warping fields. To condition our network on the target size (or the aspect-ratio), we first resize the input image $I$ to $I_R$ with the target size $H' \times W'$, and pass both $I$ and $I_R$ to the encoder, yielding two feature maps $F, F_R$:

$$F = \text{Encoder}(I), F_R = \text{Encoder}(I_R). \tag{2}$$

We notice the resized input $I_R$ already provides the coarse position of each object at the target size, but with a distorted structure. The input image $I$, however, has undistorted structure but no knowledge about the positions at the target size. We thus leverage the cross-attention blocks (Weinzaepfel

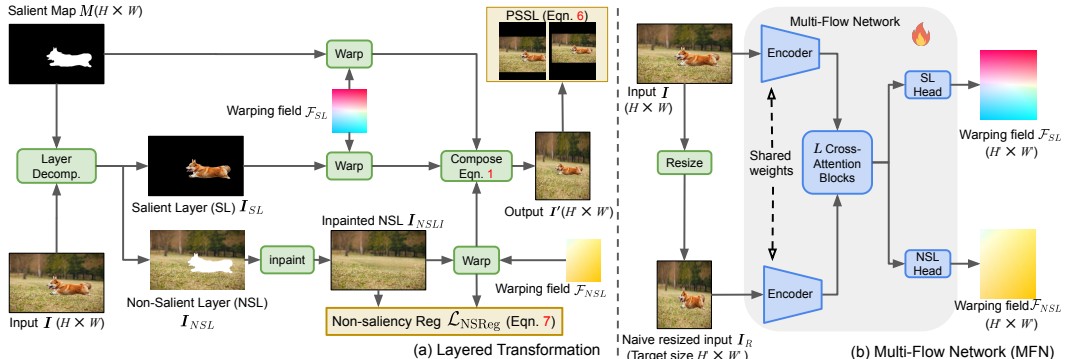

Figure 3: **Overview of HALO.** We retarget an input image $I \in \mathbb{R}^{H \times W}$ to an output image $I'$ at the target size $H' \times W'$. (a) **Layered Transformation.** We decompose the input image into a salient layer (SL) $I_{SL}$ and a non-salient layer (NSL) $I_{NSL}$ with a saliency map from (Gao et al., 2024). We inpaint the hole in $I_{NSL}$ by (Suvorov et al., 2022) to obtain the inpainted NSL $I_{NSLI}$. We then transform $I_{SL}$ and $I_{NSLI}$ with the predicted warping fields $\mathcal{F}_{SL}$ and $\mathcal{F}_{NSL}$, respectively. We also warp the saliency map $M$ with $\mathcal{F}_{SL}$ to obtain a warped saliency map $M'$. We obtain the output $I'$ by composing the warped layers with $M'$ via Eqn. 1. To train our model, we use our Perceptual Structure Similarity Loss (PSSL, Eqn. 6) and non-saliency regularization (Eqn. 7). (b) **Multi-Flow Network.** Our Multi-Flow Network (MFN) takes the input image $I \in \mathbb{R}^{H \times W}$ and its resized version $I_R \in \mathbb{R}^{H' \times W'}$ as input. $I$ and $I_R$ are encoded with a shared encoder. The resulting feature maps are then passed into $L$ cross-attention blocks. Finally, Salient-Layer (SL) head and Non-Salient Layer (NSL) head predict a salient flow $\mathcal{F}_{SL}$ and a non-salient flow $\mathcal{F}_{NSF}$ for the corresponding layers.

et al., 2023) to exchange the information between $I$ and $I_R$. Inspired by previous work (Granot et al., 2022) where each patch in the resized image queries the key patches in the input image via a non-differentiable nearest neighbor search, we adapt this idea and make it differentiable with a cross-attention mechanism. We consider the resized feature $F_R$ as query, the input feature $F$ as both key and value, and apply $L$ cross-attention blocks to them to obtain the output feature map:

$$O = \underbrace{\text{CrossAttn}_L \circ \cdots \circ \text{CrossAttn}_1(F_R, F)}_{L \text{ blocks}} . \qquad (3)$$

Finally, two heads predict two vector fields $\mathcal{F}_{SL}, \mathcal{F}_{NSL} \in \mathbb{R}^{H' \times W' \times 2}$ for warping in Eqn. 1:

$$\mathcal{F}_{SL} = \text{Head}_{SL}(O), \mathcal{F}_{NSL} = \text{Head}_{NSL}(O) , \qquad (4)$$

where $\mathcal{F}_{SL}$ is for salient layer and $\mathcal{F}_{NSL}$ for non-salient layer. Please refer to the **Supplementary Material** for more details.

## 3.3 PERCEPTUAL STRUCTURE SIMILARITY LOSS

One of the challenges of training an image retargerting model is the absence of paired data for supervision. Previous works, such as (Cho et al., 2017b; Tan et al., 2019; Mastan & Raman, 2020) use a perceptual loss (*e.g.,* VGG loss (Simonyan et al., 2014) or LPIPS (Zhang et al., 2018)) between the input and the output as a weak supervision. These perceptual loss functions calculate the distance between feature maps via a pretrained network, and do not enforce a strict supervision as pixelwise $\ell_1$ or $\ell_2$ losses. However, popular perceptual losses like LPIPS are less sensitive to structural distortions compared to DreamSim (Fu et al., 2023) in Figure 4. Therefore, we adopt DreamSim as our perceptual quality metric.

Unfortunately, directly using DreamSim does not work for image retargeting, since DreamSim is trained on square, undistorted images and preprocesses images by resizing them to a fixed square size $224 \times 224$. As shown in Figure 5, the preprocessed $I_R$ (at $224 \times 224$) exhibits a very small Dream loss with the input image $I$, despite $I_R$ having distortion at the target size. Consequently, supervising the training with DreamSim loss between the input $I$ and the output leads to a similar,

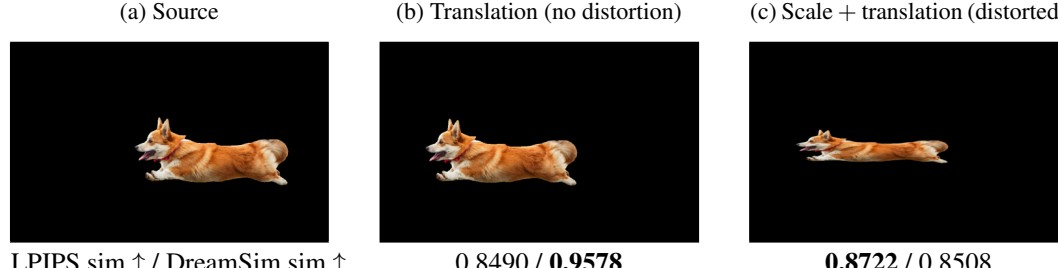

| (a) Source | (b) Translation (no distortion) | (c) Scale + translation (distorted) |
|---|---|---|
| LPIPS sim.↑ / DreamSim sim.↑ | 0.8490 / **0.9578** | **0.8722** / 0.8508 |

Figure 4: **Comparison between DreamSim and LPIPS.** We calculate the similarities of the features from LPIPS (Zhang et al., 2018) and DreamSim (Fu et al., 2023) for image pairs (a, b) and (a, c), and report the results under each column (LPIPS sim.↑ / DreamSim sim.↑). Surprisingly, the distorted result in (c) shows a higher LPIPS similarity to the source image compared to the undistorted image in (b). DreamSim, however, is more sensitive to structural similarity, showing a higher score for the undistorted image pair (a, b) and a lower score for the distorted pair (a, c).

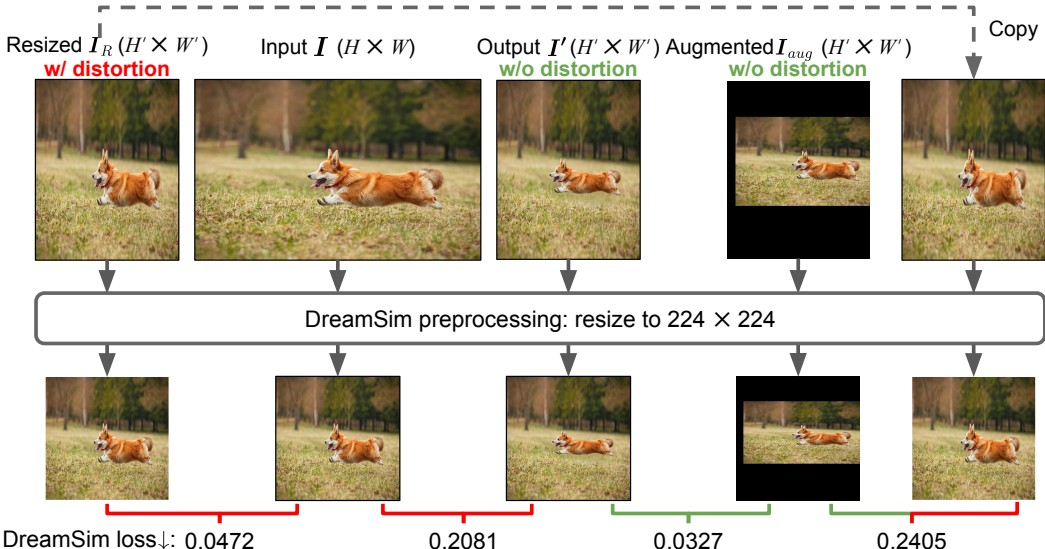

Figure 5: **Layout Augmentation.** Because DreamSim (Fu et al., 2023) preprocesses the images by resizing them to $224 \times 224$, after preprocessing, the naively resized input $\boldsymbol{I}_R$ (distorted at the target size $H' \times W'$) and the input $\boldsymbol{I}$ have a similar structure and result in a small DreamSim loss. On the other hand, the layout augmentation $\boldsymbol{I}_{aug}$ (undistorted at the target size) has a small DreamSim loss with the (ideally) undistorted output $\boldsymbol{I}'$. Therefore, to obtain an undistorted output, we compute the DreamSim loss between the output $\boldsymbol{I}'$ and $\boldsymbol{I}_{aug}$ as supervision, instead of between $\boldsymbol{I}'$ and $\boldsymbol{I}$.

distorted output as $\boldsymbol{I}_R$ at the target size. This makes the original DreamSim loss not suitable for image retargeting.

To adapt DreamSim to image retargeting, we propose to apply a random layout transformation (with scaling $s$ and translation $\boldsymbol{t}$) to disturb the input $\boldsymbol{I}$ at the target size $H' \times W'$ as an augmentation.

$$\boldsymbol{I}_{aug} = \text{Warp}(\boldsymbol{I}, \mathcal{F}(s, \boldsymbol{t})), \tag{5}$$

where $\boldsymbol{I}_{aug} \in H' \times W'$, and the warping field $\mathcal{F}(s, t) \in \mathbb{R}^{H' \times W' \times 2}$ is determined by the scaling factor $s$ and a 2D translation $\boldsymbol{t} = [t_1, t_2]$ both drawn from uniform distributions. This results in images $\boldsymbol{I}_{aug}$ *without distortions* at target size $H' \times W'$ as shown in Figure 3 and Figure 5. We encourage readers to refer to the **Supplementary Material** for more examples from the layout augmentation. We use $\boldsymbol{I}_{aug}$ as a pseudo ground truth and leverage DreamSim's structure-awareness as supervision during training and denote **Perceptual Structure Similarity Loss** (PSSL) as

$$\mathcal{L}_{PSSL}(\boldsymbol{I}', \boldsymbol{I}) = \mathcal{L}_{\text{DreamSim}}(\boldsymbol{I}', \boldsymbol{I}_{aug}). \tag{6}$$

### 3.4 TRAINING LOSS

**PSSL.** As described in Section 3.3, we use PSSL as our main training loss. We also study the popular LPIPS (Zhang et al., 2018) and demonstrate that DreamSim (Fu et al., 2023) works better than the LPIPS loss for the image retargeting (See Figure 7 and Table 3).

**Non-saliency regularization.** We further observe that, although the layered transformations (Eqn. 1) significantly mitigate the OOB issue, some extreme cases still yield OOB artifacts (See Figure 7, w/o $\mathcal{L}_{\text{NSReg}}$). The OOB issue primarily comes from the inpainted non-salient layer $\boldsymbol{I}_{NSLI}$. We use a pixelwise $\ell_2$ loss between the warped inpainted non-salient layer and the original one to encourage a mild transformation:

$$\mathcal{L}_{\text{NSReg}} = \frac{1}{N_{\text{pixel}}}||\boldsymbol{I}_{NSLI} - \text{Resize}(\text{Warp}(\boldsymbol{I}_{NSLI}, \mathcal{F}_{NSL}))||_2 \,, \tag{7}$$

where we resize the warped inpainted non-salient layer to the same size of $\boldsymbol{I}_{NSLI}$, and $N_{\text{pixel}}$ is the total number of pixels in $\boldsymbol{I}_{NSLI}$.

**Total loss.** Our training loss is

$$\mathcal{L}_{\text{total}} = \mathcal{L}_{PSSL} + \lambda_{\text{NSReg}}\mathcal{L}_{\text{NSReg}} \,, \tag{8}$$

where PSSL serves as our main loss, $\mathcal{L}_{\text{NSReg}}$ is a non-saliency regularization regularization term, and $\lambda_{\text{NSReg}}$ controls the strength of $\mathcal{L}_{\text{NSReg}}$. In practice, we use $\lambda_{\text{NSReg}} = 2.0$.

## 4 EXPERIMENTAL RESULTS

### 4.1 SETUP

**Dataset.** We train our model on the UHRSD dataset (Xie et al., 2022), which consists of $4,932$ training images and $988$ test images. Each image comes with an annotated saliency map. It covers diverse image categories including natural landscapes, street views, and animals. During training, we resize the images so that their shorter side is scaled to $512$. For example, if the height is greater than the width, the image is rescaled to $(512 \times \frac{\text{height}}{\text{width}}, 512)$. We group the images by their aspect ratios and sample images from the same group into each batch. We test our model and compare with other baseline approaches on the common RetargetMe (Rubinstein et al., 2010) benchmark. RetargetMe contains 80 images with different scaling factors ($0.50, 0.75$ and $1.25$) for the test.

**Evaluation metrics.** Previous evaluations (Cho et al., 2017b; Tan et al., 2019; Kajiura et al., 2020) on image retargeting have relied heavily on user studies. Given the rapid advancements of the recent visual representation learning, we propose to use pretrained networks to predict the image features and assess the quality of the outputs based on these features. We use CLIP image embeddings (Radford et al., 2021) for the **content** evaluation. We compute the similarity between the input image embedding and the output image embedding. To assess **structure** consistency, we use DreamSim similarity (Fu et al., 2023), which focuses on mid-level differences such as structure and layout. We use the original DreamSim since we do not wish introducing randomness from Eqn. 5 into evaluations. We use MUSIQ score (Ke et al., 2021) for **aesthetics** evaluation. To better align with other metrics, such as DreamSim and CLIP similarity, we re-normalize the MUSIQ score as a percentage. Image retargeting also requires to minimize **visual artifacts**, such as object distortion, missing or duplicated contents, or OOB artifacts. Since current assessment models struggle to reliably detect these artifacts, we conduct a user study where participants select the output with the best image quality. We use user preferences across different methods as a metric for visual artifact evaluation. We include details about our user study in the **Supplementary Material**.

### 4.2 IMPLEMENTATION DETAILS

**Model.** For the encoder of our MFN, we adopt the same CNN-based encoder as in (Peebles et al., 2022). We then use $L = 3$ cross-attention blocks. For each output head, we predict an affine transformation matrix and convert it into a sampling grid. Please refer to our **Supplementary Material** for details.

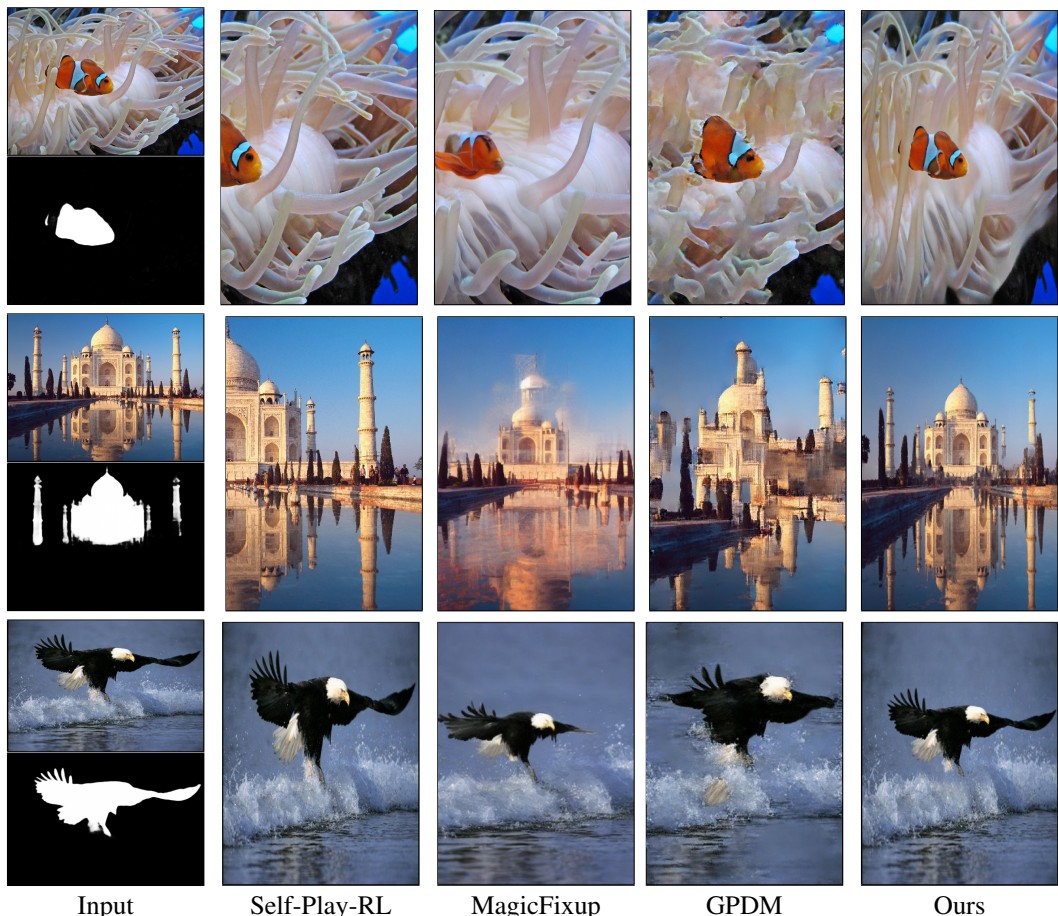

| Input | Self-Play-RL | MagicFixup | GPDM | Ours |

Figure 6: **Qualitative comparison.** We compare our method with state-of-the-art image retargeting methods: Self-Play-RL (Kajiura et al., 2020), MagicFixup (Alzayer et al., 2024), GPDM (Elnekave & Weiss, 2022). We show the input image and its saliency map from (Gao et al., 2024) in the first column. Our model preserves the structure and the content of the input images. Notably, in the "fish" case, our model is aware of the *affordance* between the fish and the sea anemone.

**Hyperparameters.** We train our network with an initial learning rate $\alpha = 1 \times 10^{-4}$ and an Adam optimizer (Kingma & Ba, 2015). The learning rate decays by a factor of $0.9$ in every $1000$ iterations. We use a batch size of 32 and train the model for 200 epochs. During the training, we sample a random target factor from $\{0.50, 0.75, 1.25, 1.50\}$ for each batch. We then randomly choose to change the height or the width of the image with the sampled factor for the current batch. For example, if we choose to change width and pick a factor $0.50$, we aim to change images' width to its half in this batch. We train our model with 2 NVIDIA A100 40G GPUs for around 2 days.

## 4.3 COMPARISON WITH PREVIOUS METHODS

We compare with three different lines of works:

- Overfitting via a generative model, including SINE (Zhang et al., 2023), SinDDM (Kulikov et al., 2023), GPDM (Elnekave & Weiss, 2022) and GPNN (Granot et al., 2022);
- Feed-forward approaches including Self-Play-RL (Kajiura et al., 2020), Cycle-IR (Tan et al., 2019) and WSSDCNN (Cho et al., 2017b).
- Drag-style editing methods. We first place the input at the center of a black canvas with the target size, and then outpaint the boundary with LAMA (Suvorov et al., 2022) if necessary. Finally we use a drag editing method to adjust the scale and the location of the salient

objects with a mask from the saliency detector (Gao et al., 2024). The scaling factor is calculated by $\frac{H'W'}{HW}$, and the translation by the shift of the centroid of the saliency mask. We compare with two state-of-the-art drag editing methods, MagicFixup (Alzayer et al., 2024) and DragDiffusion (Mou et al., 2024).

**User study.** We also conduct a user study among 16 participants on all 80 images (1280 votes) in RetargetMe (Rubinstein et al., 2010). We report the results in Table 1. Our model achieves significantly higher user preference compared to other methods. This indicates that our method aligns more closely with the human perception than other methods.

Table 1: **User study.** Our method HALO is preferred by users by a large margin.

|  | User preference (%) |
| --- | --- |
| GPDM (Elnekave & Weiss, 2022) | 5.47 |
| Self-Play-RL (Kajiura et al., 2020) | 20.86 |
| MagicFixup (Alzayer et al., 2024) | 30.23 |
| HALO (Ours) | **43.44** |

**Quantitative evaluation.** We report quantitative evaluation results in Table 2. Our method achieves the best performance in terms of content and structure preservation. While it performs slightly worse than Self-Play-RL on aesthetics, our model outperforms all others when averaging across all three metrics, yielding the highest overall score. Notably, compared to optimization-based generative models, our approach enjoys faster inference speed while achieving superior performance.

Table 2: **Quantitative comparison.** We compare our method with different types of methods, including generative modeling (*e.g.*, SINE), feed-forward prediction (*e.g.,* Cycle-IR) and drag-style editing (*e.g.,* DragDiffusion), on the RetargetMe dataset (Rubinstein et al., 2010). The test-time runtime for each method is measured on a $1024 \times 813$ image using a single NVIDIA A100 GPU. We compute the CLIP (Radford et al., 2021) embedding similarity to measure content similarity, DreamSim (Fu et al., 2023) to measure structure similarity, and MUSIQ (Ke et al., 2021) to measure aesthetics, and report the average value across all three metrics.

|  |  | Content | Structure | Aesthetics |  |
| --- | --- | --- | --- | --- | --- |
|  | Runtime(s.) ↓ | CLIP sim.(%)↑ | DreamSim sim.(%) ↑ | MUSIQ(%)↑ | Average |
| SINE (Zhang et al., 2023) | 4550.0 | 53.3 | 59.6 | 49.2 | 54.0 |
| SinDDM (Kulikov et al., 2023) | 17424.0 | 79.1 | 40.1 | 36.0 | 51.7 |
| GPDM (Elnekave & Weiss, 2022) | 61.7 | 53.6 | 65.5 | 48.5 | 55.9 |
| GPNN (Granot et al., 2022) | 21.3 | 88.5 | 77.5 | 50.7 | 72.3 |
| Self-Play-RL (Kajiura et al., 2020) | 1.30 | 88.7 | 76.2 | **52.1** | 72.4 |
| Cycle-IR (Tan et al., 2019) | 1.01 | 86.7 | 77.0 | 50.4 | 71.4 |
| WSSDCNN (Cho et al., 2017b) | 0.79 | 85.4 | 69.6 | 41.8 | 65.6 |
| MagicFixup (Alzayer et al., 2024) | 11.0 | 84.8 | 70.1 | 47.1 | 67.3 |
| DragonDiffusion (Mou et al., 2024) | 17.5 | 89.4 | 66.8 | 51.1 | 69.1 |
| HALO (Ours) | **0.59** | **90.2** | **78.0** | 51.5 | **73.2** |

**Qualitative comparison.** We showcase some visual comparison in Figure 6. We encourage the readers to view the **Supplementary Material** for more results. Compared to overfitting generative models (Granot et al., 2022; Elnekave & Weiss, 2022), our method better preserves content and structure of the input image. Compared to other feed-forward approaches (Kajiura et al., 2020), our method introduces fewer distortions, as demonstrated in the 'eagle' example. Self-Play-RL fails to preserve the some content as shown in the "fish" and the "Taj Mahal" examples. Interestingly, our model also emerges with an understanding of "affordance"—the ability to place the salient objects appropriately. In the "fish" example, our model is the only one that successfully positions the fish behind the sea anemone, maintaining the original spatial relationships. We encourage readers to refer to our **Supplementary Material** for more comparison results and insights into the model's affordance-awareness.

Table 3: **Ablation study.** We study the effect of different components. With a single transformation, the model achieves a lower DreamSim error, yet it has OOB issue as shown in Figure 7.

| | CLIP sim.(%)↑ | DreamSim sim.(%)↑ | MUSIQ(%)↑ |
|---|---|---|---|
| Single Transformation | 88.33 | **80.8** | 47.9 |
| w/o $\mathcal{L}_{\text{NSReg}}$ | 83.60 | 77.3 | 45.3 |
| w/o augmentation | 89.69 | 76.9 | 48.9 |
| Ours (w/ LPIPS) | 89.67 | 76.9 | 49.2 |
| Ours (w/ DreamSim) | **90.17** | 78.1 | **51.5** |

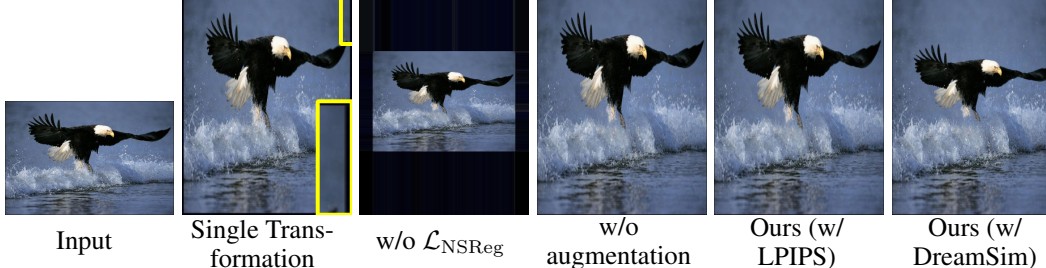

| Input | Single Transformation | w/o $\mathcal{L}_{\text{NSReg}}$ | w/o augmentation | Ours (w/ LPIPS) | Ours (w/ DreamSim) |
|---|---|---|---|---|---|

Figure 7: **Ablation study.** We show the effect of each component by removing one component each time. **With a single transformation**, it yields out-of-boundary (OOB) artifacts (such as in yellow boxes ), as the model has difficulty dealing with both the foreground and the background. **Without** $\mathcal{L}_{\text{NSReg}}$, the model also introduces OOB artifacts. **Without layout augmentation**, the model also predicts distorted results. **With LPIPS loss** (Zhang et al., 2018), the model predicts distorted results. **Our full model using DreamSim** (Fu et al., 2023) predicts results with less distortion and avoids OOB artifacts thanks to the compositional transformations.

## 4.4 ABLATION STUDY

To examine the effect of each proposed component, we conduct an ablation study. We remove one component in our full method each time, and show the results in Table 3 and Figure 7. **With one single transformation**, the model achieves the best performance on structure preservation, but it introduces OOB artifacts as shown in Figure 7. **Removing the background regularization term** $\mathcal{L}_{\text{NSReg}}$ also introduces some OOB artifacts as shown in Figure 7. **Layout augmentation** brings significant improvement for the distortions, as shown in Table 3 and Figure 7. Finally, by **replacing DreamSim with LPIPS** (Zhang et al., 2018), the model still suffers from the distorted content, further highlighting DreamSim's effectiveness in maintaining layout and structure awareness..

## 4.5 RESULTS ON IN-THE-WILD DATA

To demonstrate the generalizability of our model, we test our model on 400 in-the-wild images from Unsplash (Unsplash, 2020). We show qualitative results in Figure 8. *Without any finetuning*, our model generalizes well to diverse scenarios, varying from common objects, natural landscapes, and animals. We show more results on in-the-wild data in the **Supplementary Material**.

## 4.6 LIMITATIONS

Our current approach also has limitations. As shown in Figure 9, HALO struggles when the saliency detector (Gao et al., 2024) fails to associate the soccer ball with its shadow. We can either use a more accurate mask (*e.g.,* from (Liu et al., 2023)) or use an object association method (Alzayer et al., 2024; Winter et al., 2024) to improve the result.

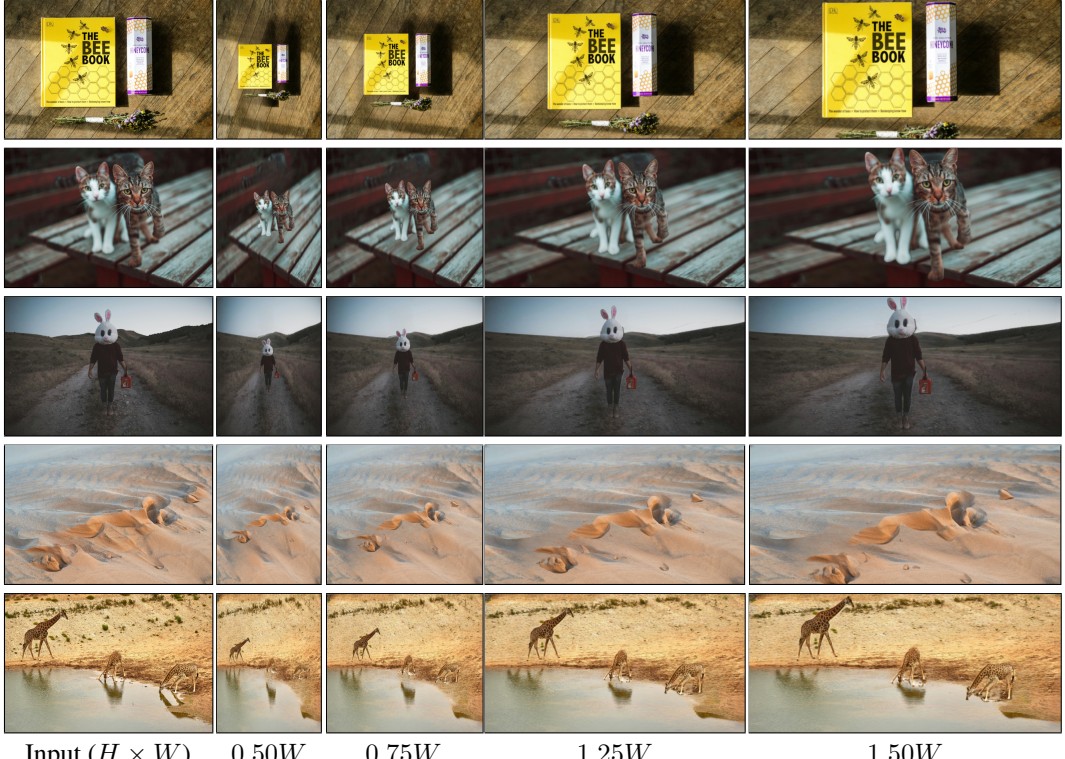

Input ($H \times W$)     0.50$W$     0.75$W$     1.25$W$     1.50$W$

Figure 8: **Qualitative results on the in-the-wild images.** *Without further finetuning*, our model generalizes to the in-the-wild images, covering common objects and animals. It works for single object and multiple objects. The input images are from Unsplash (Unsplash, 2020).

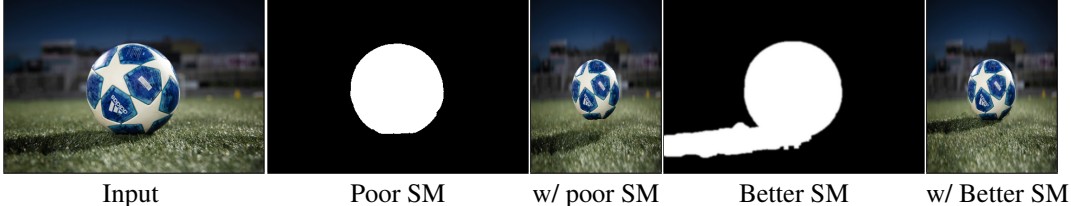

Input          Poor SM     w/ poor SM     Better SM     w/ Better SM

Figure 9: **Limitations.** Our model faces challenges with poor saliency map (SM) prediction. In this example, the saliency detector of Gao et al. (2024) fails to associate the shadow with the soccer ball, resulting in a "floating" ball. By using an improved mask that includes the shadow, our model yields a more reasonable output. We reduce the width of the image to its half in this case.

## 5 CONCLUSION

We present HALO, an end-to-end framework for image retargeting that aligns with human perception. By using a layered representation for the input and applying distinct transformations to salient and non-salient regions, our approach produces results with fewer visual artifacts, such as the OOB issue. We also introduce a new Perceptual Structure Similarity Loss (PSSL) enabling training without paired data for image retargeting and equips the model with distortion-awareness capabilities. We conduct extensive evaluations across various methods, demonstrating that HALO outperforms previous approaches. A user study further confirms that HALO aligns closely with human perception, outperforming the SOTAs by a large margin.

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

# A SUPPLEMENTARY MATERIAL

## A.1 IMPLEMENTATION DETAILS

### A.1.1 NETWORK ARCHITECTURE

**Encoder.** We use the same encoder as in GANGealing (Peebles et al., 2022). The encoder follows the architecture of the StyleGAN2 discriminator (Karras et al., 2020), with a ResNet backbone (He et al., 2016b). In practice, we use the same encoder for both the original input image and its naively resized version to obtain two feature maps. Two feature maps are fed into $L$ Cross-Attention blocks.

**Cross-Attention blocks.** To condition the network on the target image size, we choose to naively resize the input image to the target size. To better understand the rough layout at the target size, and to introduce a differentiable analogy to PNN methods (Granot et al., 2022; Elnekave & Weiss, 2022), we choose to use cross-attention mechanism to share the information between the original input and the resized input. We adopt the decoder block from CroCo-v2 (Weinzaepfel et al., 2023), where it consists of LayerNorm, SelfAttention, CrossAttention and MLP. In practice, we use $L = 3$ decoder blocks, and each block has 4 heads.

**Heads.** We use two heads for the foreground and the background, respectively. Each head predicts an affine transformation. Unlike GANGealing (Peebles et al., 2022) and NeuralGealing (Ofri-Amar et al., 2023), which compose a similarity transformation with an unconstrained flow field, we find the flow field introduces unnatural distortions so we end up without using the flow field. In practice, each head is equipped with a Linear layer to predict 5 parameters $o_1, o_2, o_3, o_4, o_5$. We construct the affine matrix $\mathbf{A}$ as follows:

$$r = \pi \cdot \tanh(o_1) \tag{9}$$

$$s_x = \exp(o_2) \tag{10}$$

$$s_y = \exp(o_3) \tag{11}$$

$$t_x = o_4 \tag{12}$$

$$t_y = o_5 \tag{13}$$

$$\mathbf{A} = \begin{bmatrix} s_x \cdot \cos(r) & -s_y \cdot \sin(r) & t_x \\ s_y \cdot \sin(r) & s_x \cdot \cos(r) & t_y \\ 0 & 0 & 1 \end{bmatrix} \tag{14}$$

To warp the image, we apply $\mathbf{A}$ to an identity sampling grid, and then apply the transformed sampling grid to the input image.

### A.1.2 PERCEPTUAL STRUCTURE SIMILARITY LOSS

We apply a random transformation to the input image as an undistorted, pseudo ground truth during the training. The transformation includes a scaling $s$ and a translation $\mathbf{t} = [t_1, t_2] \in \mathbb{R}^2$. Suppose the input image has a size of $H \times W$, and the target size is $H' \times W'$, we construct a transformation matrix $\mathbf{D}$ as follows:

$$\mathbf{D} = \begin{bmatrix} s \cdot k_x & -s \cdot k_y & t_1 \cdot H' \\ s \cdot k_y & s \cdot k_x & t_2 \cdot W' \\ 0 & 0 & 1 \end{bmatrix}, \tag{15}$$

where $k_x = \frac{H'}{H}$, $k_y = \frac{W'}{W}$. To obtain the warped image, we apply $\mathbf{D}$ to an identity sampling grid, and then apply the transformed sampling grid to the input image. In practice, we sample $s$ from a uniform distribution $\mathcal{U} \sim [0.9, 1.5]$ and $t_1, t_2$ from $\mathcal{U} \sim [-0.01, 0.01]$.

We show some examples of the random augmented images in Figure 10.

### A.1.3 TRAINING DETAILS

We train our network with an initial learning rate $\alpha = 1 \times 10^{-4}$ and an Adam optimizer (Kingma & Ba, 2015). The learning rate decays by a factor of 0.9 in every 1000 iterations. To facilitate batch training, we split the images with same aspect-ratio into different groups. At each iteration, we sample a group and a batch of images from the group. We use a batch size of 32 and train the model

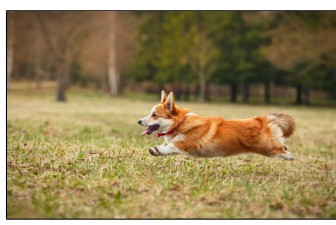 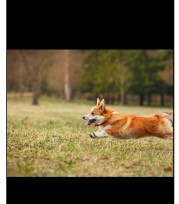 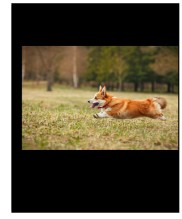 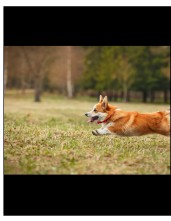

| Input | Example 1 | Example 2 | Example 3 |

Figure 10: **Examples of layout augmentation.** We show some examples of the layout augmentation. In this case, $H' = 0.5H$.

for 200 epochs. During the training, we sample a random target ratio from $\{0.50, 0.75, 1.25, 1.50\}$ at each iteration. We then randomly choose to scale the height or the width of the image with the sampled ratio factor for current batch. We train our model with 2 NVIDIA A100 GPUs for around 2 days.

## A.2 USER STUDY

We draw one method from each line of work (GPDM (Elnekave & Weiss, 2022), Self-Play-RL (Kajiura et al., 2020), MagicFixup (Alzayer et al., 2024)) and conduct a user study over 16 people. All 80 images in RetargetMe (Rubinstein et al., 2010) are evaluated. We present the input image, the output images from three other baselines, and our output image side by side, and we ask the users to select the best result based on:

- **Less distortion:** check if one image is squeezed or stretched. Choose the one has the least distortions.

- **Preserve better content:** check if one image has everything (objects, background, characters/text, etc) from the Source, or with a minimum loss of important contents.

- **Less visual artifacts:** check if one image is less sharper, has missing parts, or duplicated parts.

## A.3 ADDITIONAL RESULTS

### A.3.1 ADDITIONAL RESULTS FOR AFFORDANCE-AWARENESS

As mentioned in Figure 6 in our main paper, our model emerges an ability to understand the affordance between different objects in Figure 11. We show more results to demonstrate the understanding of affordance.

### A.3.2 ADDITIONAL COMPARISON WITH PREVIOUS METHODS

We show more results to compare with previous approaches:

- Generative model overfitting: SINE (Zhang et al., 2023), SinDDM (Kulikov et al., 2023), GPDM (Elnekave & Weiss, 2022), GPNN (Granot et al., 2022);
- Feed-forward approaches: Self-Play-RL (Kajiura et al., 2020), Cycle-IR (Tan et al., 2019), WSSDCNN (Cho et al., 2017b);
- Drag-style editing: MagicFixup (Alzayer et al., 2024), DragonDiffusion (Mou et al., 2024).

The results are shown in Figure 12 and Figure 13.

### A.3.3 ADDITIONAL RESULTS ON THE IN-THE-WILD DATA

We show more in-the-wild results in Figure 14, Figure 15 and Figure 16.

864
865
866
867
868
869
870
871
872
873
874
875
876
877
878
879
880
881
882
883
884
885
886
887
888
889
890
891
892
893
894
895
896
897
898
899
900
901
902
903
904
905
906
907
908
909

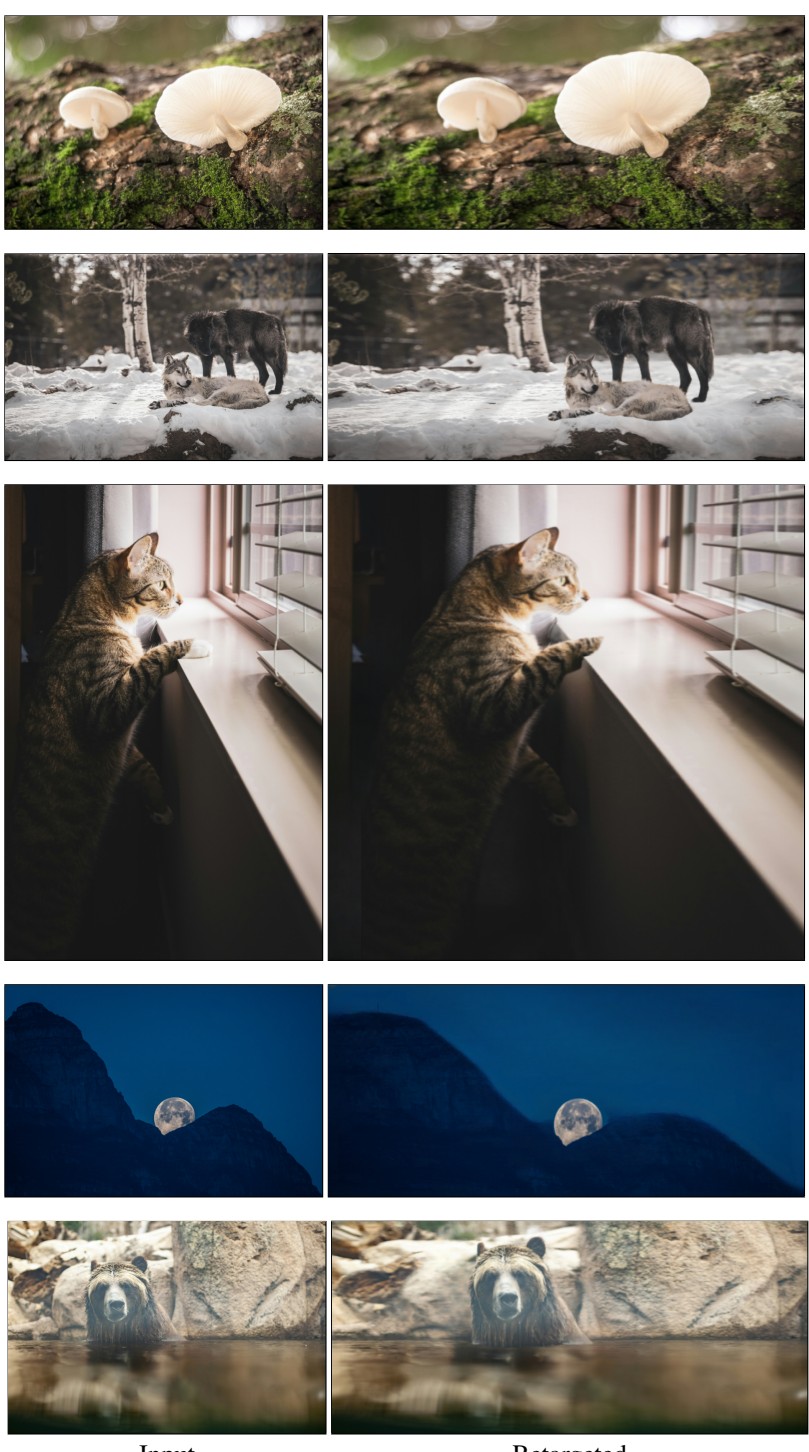

Input                              Retargeted

910
911
912
913
914
915
916
917

Figure 11: **Additional results for affordance-awareness.** Our model emerges with an ability to understand the affordance of the objects. It places the salient object properly with other objects. For example, in the "mushroom" case, mushrooms are placed near the green moss, similar to the input. In the "wolves" case, wolves are placed at a similar position as in the input image.

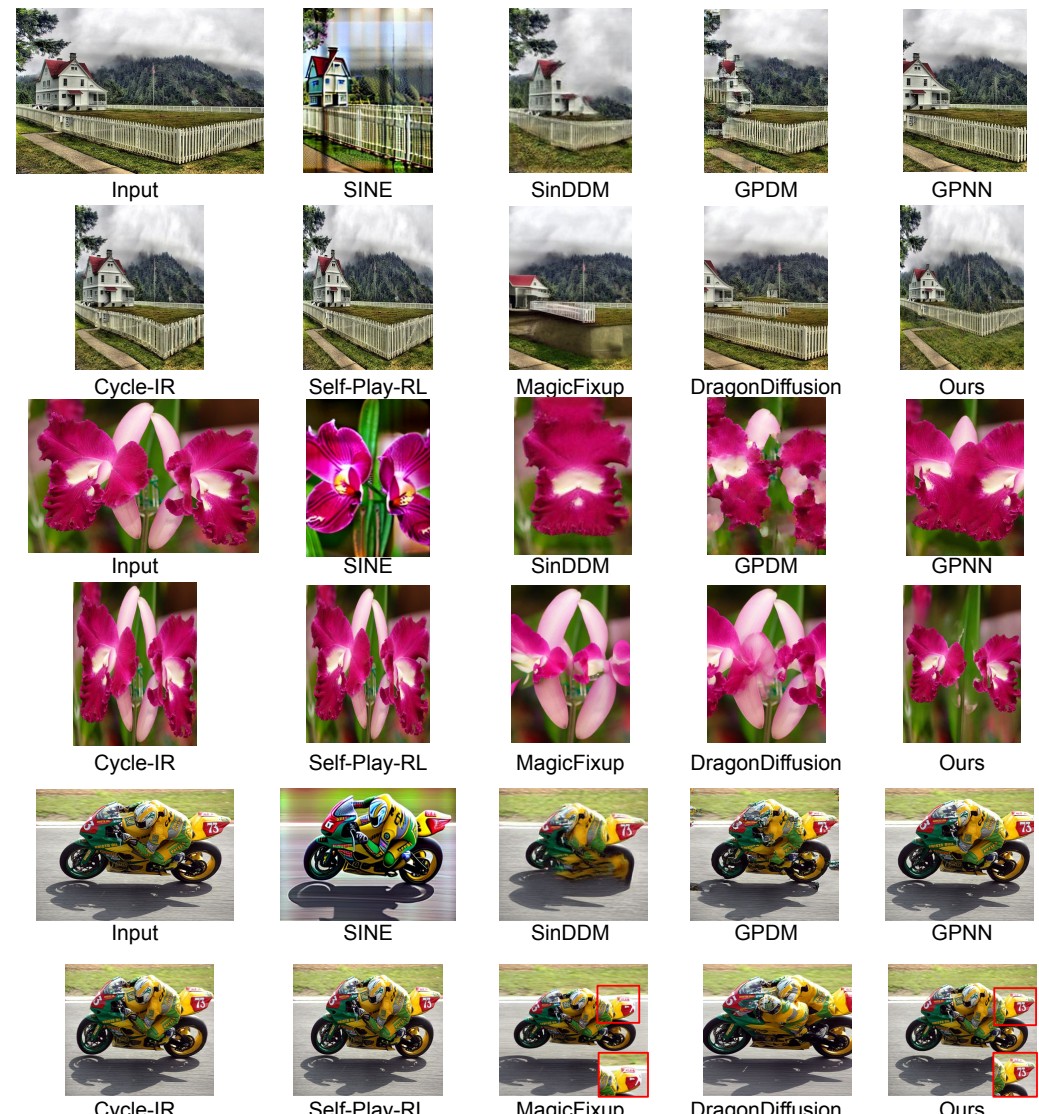

Figure 12: **Additional qualitative comparison on RetargetMe.** We show more visual comparison results. We compare with SINE (Zhang et al., 2023), SinDDM (Kulikov et al., 2023), GPDM (El-nekave & Weiss, 2022), GPNN (Granot et al., 2022), Self-Play-RL (Kajiura et al., 2020), Cycle-IR (Tan et al., 2019), WSSDCNN (Cho et al., 2017b), MagicFixup (Alzayer et al., 2024), Dragon-Diffusion (Mou et al., 2024).

### A.3.4 ADDITIONAL RESULTS WITH OTHER NO-REFERENCE METRICS

We include additional no-reference metrics in Table 4, specifically the learning-based score VILA (Ke et al., 2023) and the non-learning-based score NIQE (Mittal et al., 2012). Our HALO method demonstrates competitive performance on these no-reference metrics, achieving the highest average score.

To compute the average scores, we normalize both VILA and NIQE to percentages. For NIQE, we use $100 - \text{norm(NIQE)}$, as a lower NIQE score indicates better performance.

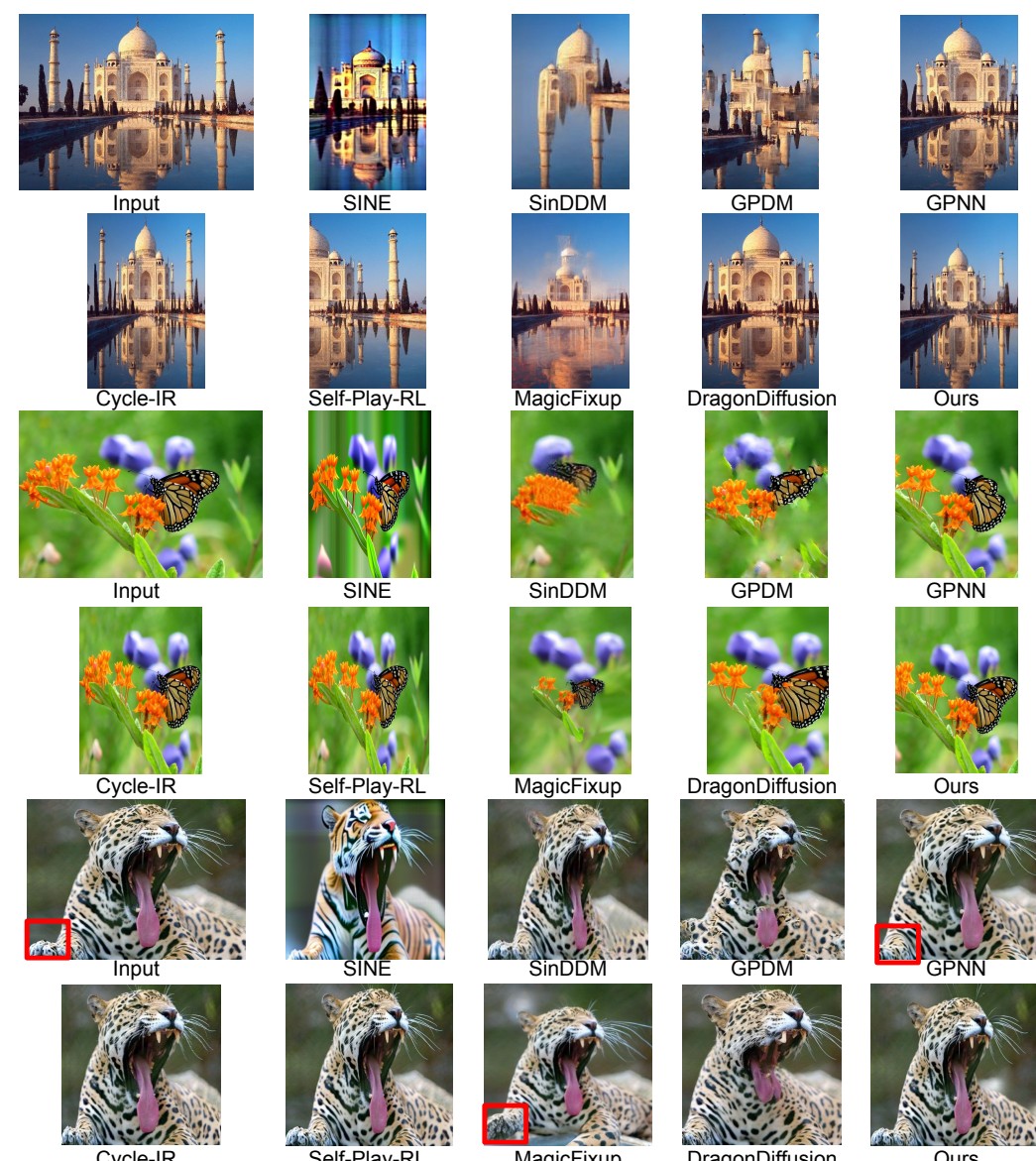

Figure 13: **Additional qualitative comparison on RetargetMe.** We show more visual comparison results. We compare with SINE (Zhang et al., 2023), SinDDM (Kulikov et al., 2023), GPDM (El-nekave & Weiss, 2022), GPNN (Granot et al., 2022), Self-Play-RL (Kajiura et al., 2020), Cycle-IR (Tan et al., 2019), WSSDCNN (Cho et al., 2017b), MagicFixup (Alzayer et al., 2024), Dragon-Diffusion (Mou et al., 2024).

## A.4 ANALYSIS OF THE OFF-THE-SHELF MODELS

### A.4.1 ANALYSIS OF THE INPAINTING MODEL

We use an off-the-shelf inpainting model, LAMA (Suvorov et al., 2022), one of the state-of-the-art image inpainting models.

**Why LAMA?** We show a qualitative comparison with another naive inpainting method from OpenCV library in Figure 17.

**If LAMA fails.** As an off-the-shelf model, LAMA could compromise when the textures are complicated. Fortunately, as shown Figure 6 and Figure 11, our model emerges with awareness of the

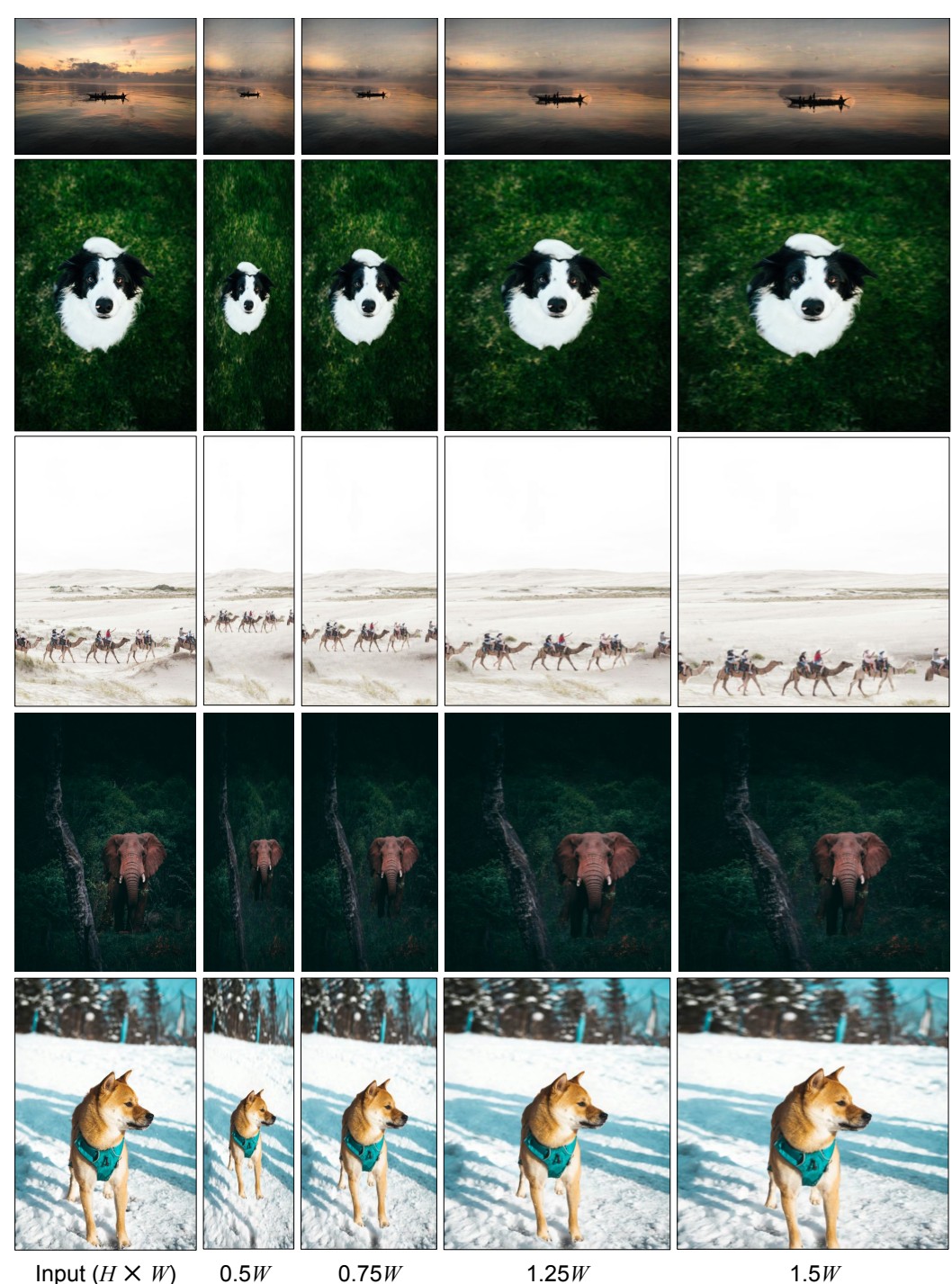

Input ($H \times W$)  0.5$W$  0.75$W$   1.25$W$    1.5$W$

Figure 14: **Additional qualitative results on the in-the-wild images.** *Without further finetuning*, our model generalizes to the in-the-wild images. The input images are from the Unsplash dataset Unsplash (2020).

affordance. It therefore places the content correctly and the undesired part is occluded. We show an example in Figure 18.

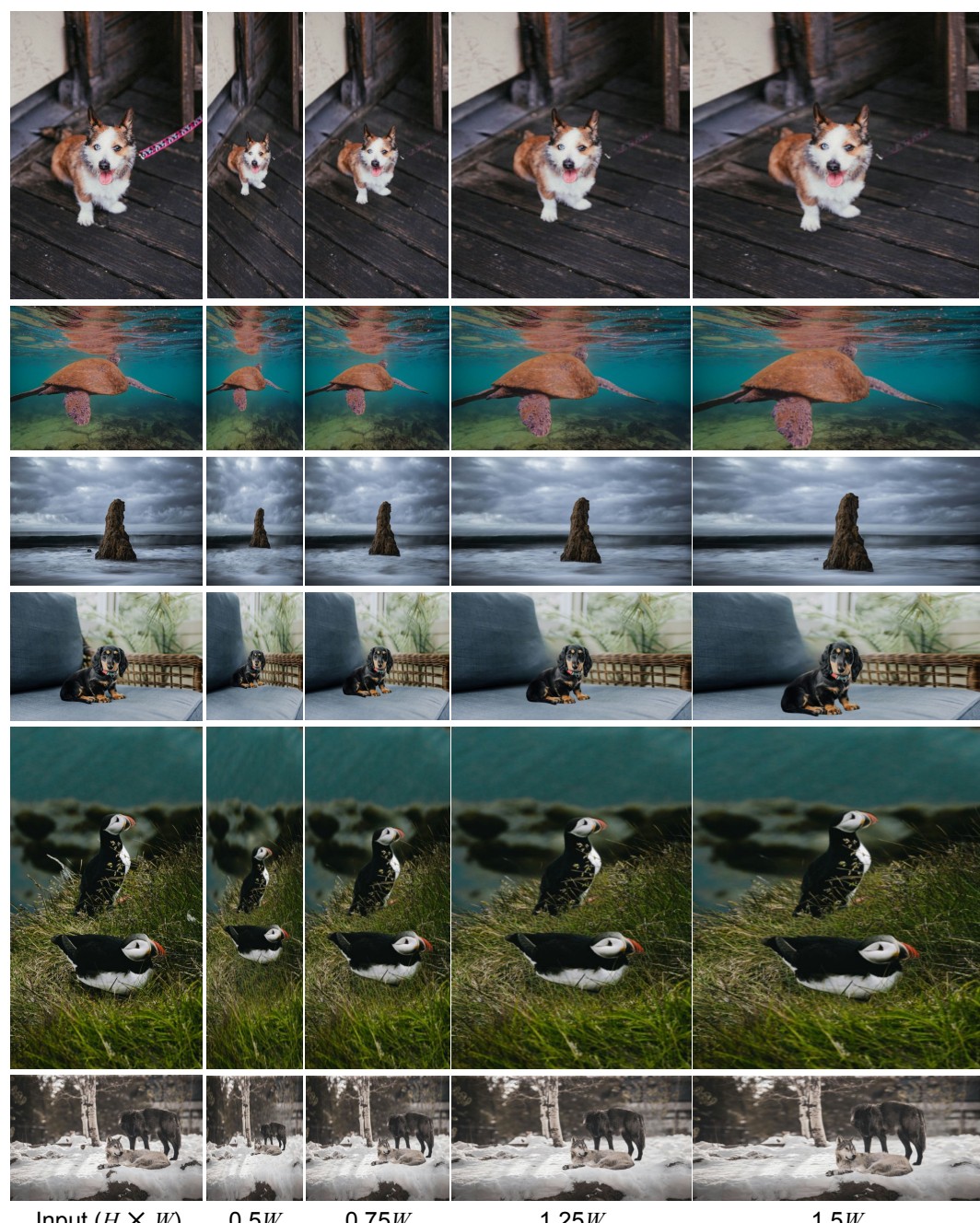

Figure 15: **Additional qualitative results on the in-the-wild images.** *Without further finetuning*, our model generalizes to the in-the-wild images. The input images are from the Unsplash dataset Unsplash (2020).

### A.4.2 ANALYSIS OF THE SALIENCY DETECTOR

We use one of the state-of-the-art saliency detectors, MDSAM (Gao et al., 2024) to predict saliency map.

**Why MDSAM?** To demonstrate the effectiveness of MDSAM, we retrain a model without MDSAM and use an all-one mask instead. We show the performance in Table 5. Without saliency detector (Gao et al., 2024), the model shows a similar result as Single Transformation. It shows a

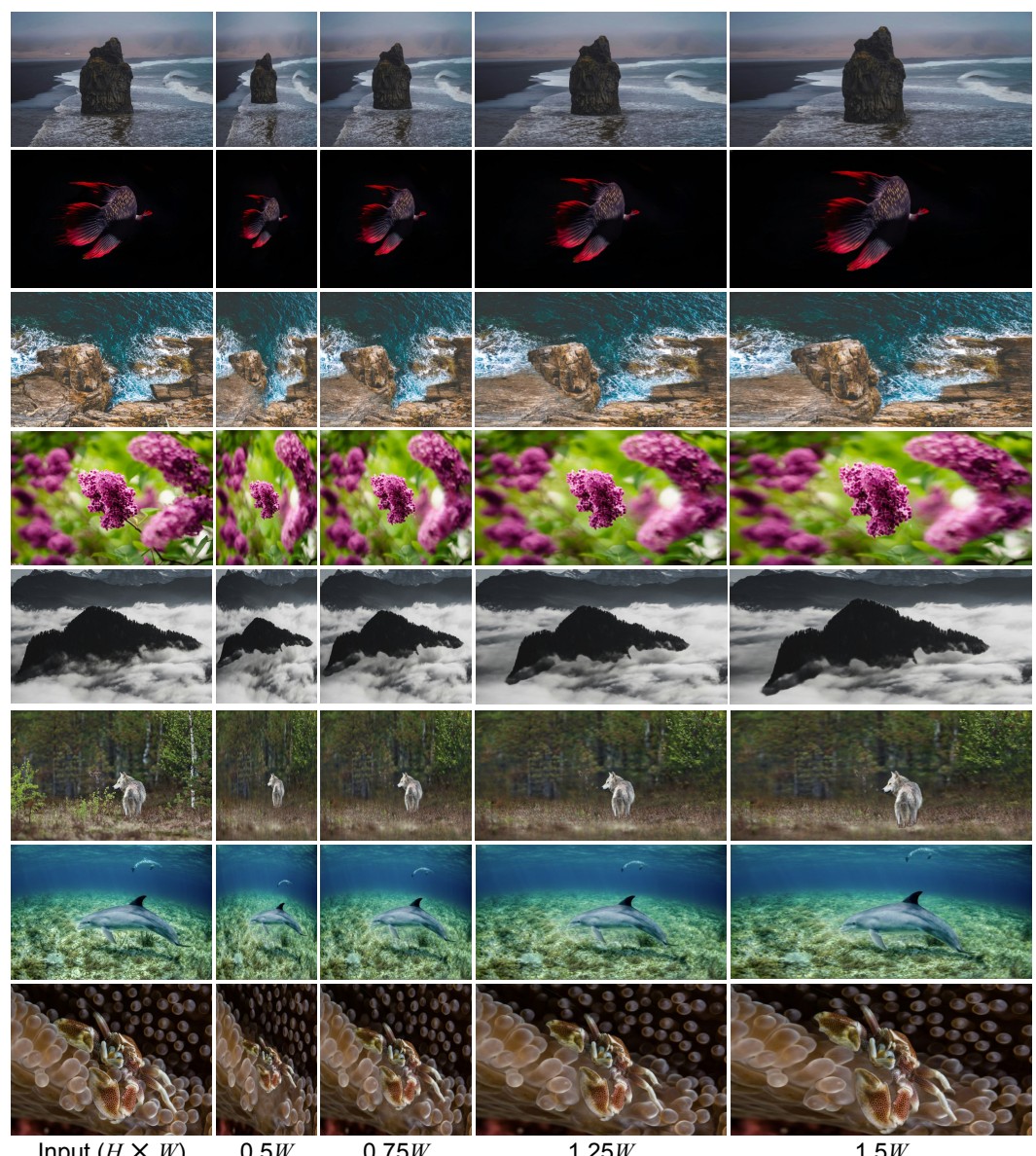

Input ($H \times W$)     0.5$W$     0.75$W$     1.25$W$     1.5$W$

Figure 16: **Additional qualitative results on the in-the-wild images.** *Without further finetuning,* our model generalizes to the in-the-wild images. In "crab" case, our method notice the "affordance" between the coral and the crab. The input images are from the Unsplash dataset Unsplash (2020).

higher DreamSim as the preprocessing of DreamSim prefers a distorted result (Figure 10). Our full model shows the highest average score over three metrics.

**If MDSAM fails.** When there are no obvious salient objects, MDSAM may produce unreliable results. In that case, we can provide the model with an all-one mask, and our model becomes a cropping model. We show an example in Figure 19. We would like to emphasize that, for this challenging case (no obvious saliency), it is ill-posed and there are multiple solutions.

## A.5 LIMITATION OF MUSIQ SCORE

We find MUSIQ (Ke et al., 2021) sometimes prefers results with distortions. We show an example in Figure 20.

Table 4: **Quantitative comparison with more no-reference scores.** We include a learning-based score VILA (Ke et al., 2023) and a non-learning-based score NIQE (Mittal et al., 2012). Our HALO method demonstrates competitive performance on these no-reference metrics, achieving the highest average score. To compute the average scores, we normalize both VILA and NIQE to percentages. For NIQE, we use $100 - \mathrm{norm}(\mathrm{NIQE})$, as a lower NIQE score indicates better performance.

| | CLIP sim.(%)↑ | DreamSim sim.(%) ↑ | MUSIQ(%)↑ | VILA(%)↑ | NIQE ↓ | Average(%)↑ |
|---|---|---|---|---|---|---|
| SINE (Zhang et al., 2023) | 53.3 | 59.6 | 49.2 | 44.5 | 5.40 | 50.5 |
| SinDDM (Kulikov et al., 2023) | 79.1 | 40.1 | 36.0 | 24.8 | 6.72 | 42.5 |
| GPDM (Elnekave & Weiss, 2022) | 53.6 | 65.5 | 48.5 | 47.3 | 4.39 | 54.2 |
| GPNN (Granot et al., 2022) | 88.5 | 77.5 | 50.7 | **50.7** | 4.74 | 64.0 |
| Self-Play-RL (Kajiura et al., 2020) | 88.7 | 76.2 | **52.1** | **50.7** | 4.40 | 64.8 |
| Cycle-IR (Tan et al., 2019) | 86.7 | 77.0 | 50.4 | 45.2 | 4.43 | 63.0 |
| WSSDCNN (Cho et al., 2017b) | 85.4 | 69.6 | 41.8 | 33.0 | 6.84 | 52.3 |
| MagicFixup (Alzayer et al., 2024) | 84.8 | 70.1 | 47.1 | 42.4 | 4.48 | 59.9 |
| DragonDiffusion (Mou et al., 2024) | 89.4 | 66.8 | 51.1 | 47.1 | **3.96** | 62.9 |
| HALO (Ours) | **90.2** | **78.0** | 51.5 | 48.1 | 4.33 | **64.9** |

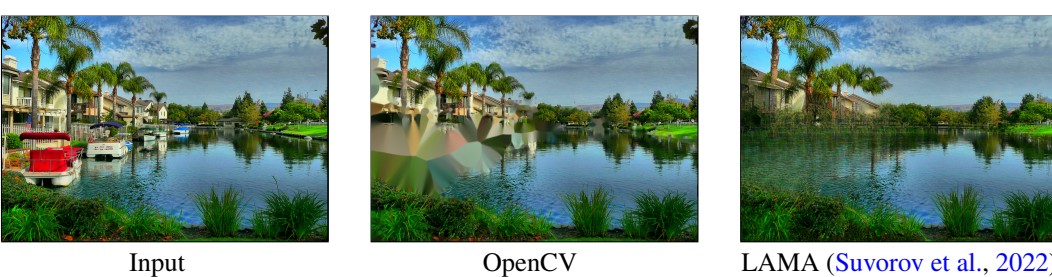

|     Input     |     OpenCV     |     LAMA (Suvorov et al., 2022)     |

Figure 17: **Why we use LAMA for inpainting (Suvorov et al., 2022).** We compare LAMA, a state-of-the-art inpaining model, with another off-the-shelf inpainting method from OpenCV. LAMA shows significantly better performance.

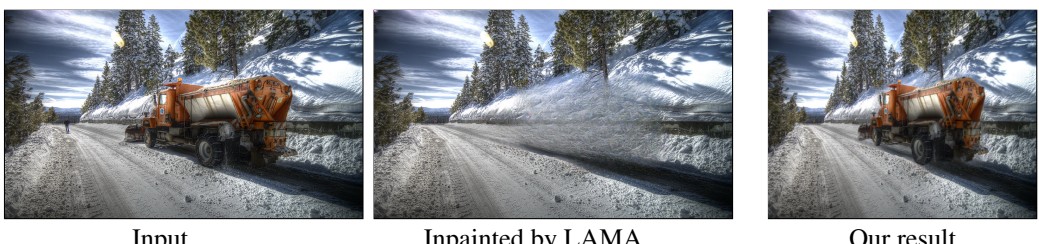

|     Input     |     Inpainted by LAMA     |     Our result     |

Figure 18: **Affordance helps LAMA.** LAMA (Suvorov et al., 2022) could fail when the inpainting mask is large. In this case, the inpainted result shows undesired textures. Fortunately, as shown in Figure 6 and Figure 11, our model emerges with awareness of the affordance. It therefore places the content correctly and the undesired part is occluded.

Table 5: **Performance without saliency detector.** Without saliency detector (Gao et al., 2024), the model shows a similar result as Single Transformation. It shows a higher DreamSim as the preprocessing of DreamSim prefers a distorted result (Figure 10). Our full model shows the highest average score over three metrics.

| | CLIP sim.(%)↑ | DreamSim sim.(%)↑ | MUSIQ(%)↑ | average |
|---|---|---|---|---|
| Single Transformation | 88.33 | **80.8** | 47.9 | 72.3 |
| w/o saliency detector | 87.25 | 82.1 | 48.6 | 72.6 |
| Ours (full) | **90.17** | 78.1 | **51.5** | **73.3** |

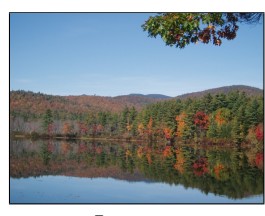 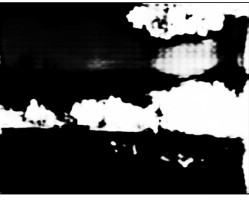 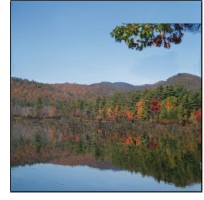 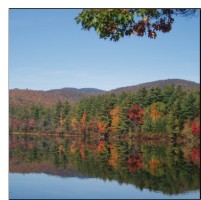

| Input | MDSAM prediction | Result w/ MDSAM | Result w/ all-one mask |

Figure 19: **All-one mask helps MDSAM.** When there are no obvious salient objects, it may produce unreliable results. In that case, we can provide the model with an **all-one** mask, and our model becomes a cropping model.

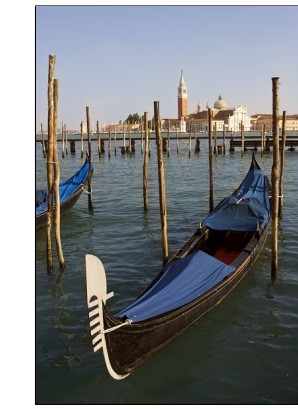 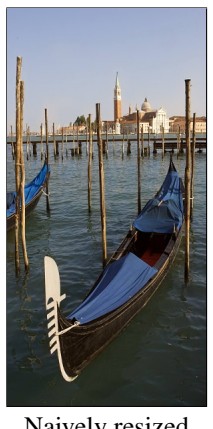 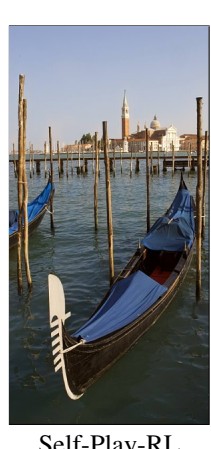 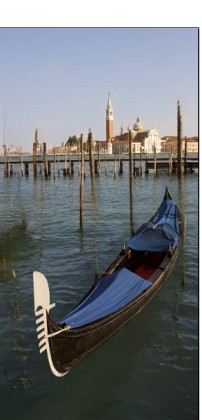

| Input | Naively resized MUSIQ↑: 53.43 | Self-Play-RL MUSIQ↑: 53.45 | HALO (ours) MUSIQ↑: 51.20 |

Figure 20: **Limitation of MUSIQ (Ke et al., 2021).** We find MUSIQ itself may *not* be sensitive to the distortions as they are trained with undistorted images. Self-Play-RL (Kajiura et al., 2020) shows a similar result to naively resized output, which has distortions. Our result, however, showing less distortions, receives a lower MUSIQ score.

## A.6 LPIPS WITH OUR AUGMENTATION

We additionally show the result using LPIPS with our proposed layout augmentation (Section 3.3) in Table 6. With our layout augmentation, the performance of LPIPS is improved, but still worse than the one with DreamSim (Fu et al., 2023), as LPIPS is not sensitive to the structure (Figure 4).

Table 6: **LPIPS with our layout augmentation.** With augmentation, LPIPS gets improved. However, its performance is still worse than DreamSim as LPIPS is not sensitive to the structure (Figure 4).

|  | CLIP sim.(%)↑ | DreamSim sim.(%)↑ | MUSIQ(%)↑ |
|---|---|---|---|
| Ours (w/ LPIPS) | 89.67 | 76.9 | 49.2 |
| Ours (w/ LPIPS + aug.) | 90.15 | 77.2 | 50.3 |
| Ours (w/ DreamSim) | 89.69 | 76.9 | 48.9 |
| Ours (w/ DreamSim + aug.) | **90.17** | **78.1** | **51.5** |

