# OpenReview forum: "HALO: Human-Aligned End-to-end Image Retargeting with Layered Transformations"
_ICLR.cc/2025/Conference — Submitted to ICLR 2025_

### Official Review · Reviewer_AZT5 · 2024-11-01

**Soundness:** 3
**Presentation:** 3
**Contribution:** 2
**Rating:** 5
**Confidence:** 5

**Summary:**

This paper presents an end-to-end trainable approach for image retargeting. The authors introduce a novel layered transformation to learn two warping fields utilizing a pre-defined saliency map and inpainting model. Additionally, they develop a tailored perceptual structural similarity loss for model training. Quantitative results and user studies showcase the method's effectiveness.

**Strengths:**

1. This paper proposes a end-to-end learnable image retargeting framework and the results are impressive.
2. This paper is well-written, showcasing a clear and coherent structure that effectively guides the reader through its main arguments and findings.

**Weaknesses:**

1. The performance of the proposed approach is fundamentally dependent on the choice of the saliency model and the image inpainting model utilized. The selected saliency model plays a crucial role in accurately identifying and highlighting the most important regions of the image. Meanwhile, the image inpainting model contributes significantly to the quality of the restored content. Together, these models work in tandem to optimize the overall results, making their selection a critical factor in achieving superior performance.
2. Although the paper offers a well-structured and clear introduction to the methodological process, guiding readers through the steps and techniques employed, it falls short in addressing the underlying principles that inform these methods.

**Questions:**

1. What image resolutions can the proposed method accommodate, and what is the processing time like?
2. What is the key to the proposed method's performance being superior to that of the comparison methods?

---

> ### Author Response · Authors · 2024-11-22
>
> Thank you for your valuable comments. We have uploaded a **revised version** with updated results and comments. Please download the updated version on **the upper-right PDF button.**
>
> We address your concerns below.
>
> **W1: Limitations of the off-the-shelf methods.**
> ---
> **A1**: We acknowledge that these off-the-shelf methods, i.e., inpainting model and saliency detector, have their limitations.
>
> [**Inpainting model**]
>
> **Why LAMA?** For our inpainting model (LAMA), we use a dilated mask to improve the inpainting quality. We find it sufficient for our task. We here show a weaker inpaiting model, `cv2.inpaint`, to show LAMA’s superiority. We show the visualization in **Figure 17** in our revised version. LAMA shows better inpainting ability.
>
> **If LAMA does not work well.** We show an example below where LAMA inpaints the non-saliency image with undesired textures. Luckily, as shown in L428-L431, Figure 6 and Figure 11, our model emerges with awareness of the affordance. It therefore places the content correctly and the undesired part is occluded. We include an example in Figure 18 in our revised version.
>
> [**Saliency detector**]
>
> **Why MDSAM?** To demonstrate the effectiveness of the saliency detector, we also report the results where we do not use the saliency detector and let the model take an all-one mask as input and predict transformations. We report its performance below (also reported in **Table 5** in our revised version). This demonstrate that the effectiveness of MDSAM.
>
> | Model | CLIP sim.(%) | DreamSim sim.(%) | MUSIQ (%) | Average |
> |----------|----------|----------|----------|----------|
> |Single Transformation   | 88.33 | 80.8 | 47.9 |72.3
> |w/o saliency detector    | 87.25     | **82.1**     | 48.6     | 72.6     |
> | Ours (full)    | **90.17**   | 78.1     | **51.5**     | **73.3**     |
>
> **If MDSAM fails:** for the saliency detector, we have demonstrated one limitation in Section 4.6. Also when there are no obvious salient objects, it may produce unreliable results. In that case, we can provide the model with an **all-one mask**, and our model becomes a cropping model. We show an example in **Figure 19** in our revised version.
> We would like to emphasize that, for this challenging case (no obvious saliency), it is ill-posed and there are multiple solutions.
>
> **W2: Principles informing the proposed techniques.**
> --
> **A2**: Thank you for your suggestions. We would like to emphasize our method section by including the following intuitions to make the transitions smoother. We are open to more discussions.
> - **Image Retargeting**. Image Retargeting requires (i) structure preservation, (ii) content preservation, and (iii) less visual artifacts.
> - **For structure preservation, we use Perceptual Structure Similarity Loss (PSSL)**. Given the absence of paired training data, many previous works use perceptual losses as a weaker supervision. However, we notice that popular loss function, such as LPIPS, are less sensitive to the structure distortion (Figure 4). We therefore propose to adopt DreamSim, which is more sensitive to the mid-level features. However, directly using DreamSim still brings problems (Figure 5) as it is trained on squared images without distortions. To address this problem, we propose a new augmentation trick for it and propose Perceptual Structure Similarity Loss (PSSL).
> - **For content preservation, we use Multi-Flow Network (MFN)**. Previous methods often predict one single transformation, which introduce out-of-boundary (OOB) artifacts (Figure 2, WSSDCNN [1]) and loss of content. To address this, we propose to use multiple transformations.
> - **By combining them together**, our HALO method achieves less **visual artifacts**.
>
> **Q1: Image resolutions and processing time**
> ---
> **A3**:
> - **Resolutions.** Our proposed method can handle arbitrary resolutions. We include in-the-wild data evaluation in Figure 8, and Figure 14, Figure 15 and FIgure 16 in the supplementary material. These in-the-wild data have diverse image resolutions.
> - **Runtime**. We include a comparison of runtimes in Table 2. Our model enjoys a faster inference speed with a runtime of 0.59s on an 1024 $\times$ 813 image on a single A100.
>
> **Q2: Key to success.**
> ---
> **A4**: We show the effectiveness of each component in Table 3. We consider the layered transformations and our proposed PSSL as the key to the improvement. With layered transformations, we bring the CLIP from 88.33 to 89.69, and MUSIQ from 47.9 to 48.9. With PSSL, we further improve CLIP from 89.69 to 90.17, and MUSIQ from 48.9 to 51.5
>
> References
> ---
> [1] Cho, Donghyeon, et al. "Weakly-and self-supervised learning for content-aware deep image retargeting." ICCV 2017.

---

> ### Author Response · Authors · 2024-11-24
>
> Hi Reviewer AZT5,
>
> Thanks again for your valuable feedback to our paper. We have added more experimental results to address your comments. Let us know if you still have concerns or not. If you think your concerns are resolved. We kindly ask you to reply to this thread and change the score accordingly. If you still have concerns, let us know, we will add more results if needed. Your reply means a lot to us and helps to build a better research community. Thanks again.

---

> > ### Author Response · Authors · 2024-11-25
> > **Please let us know whether all questions are addressed**
> >
> > Dear reviewer,
> >
> > We have provided answers and explanations to the questions of reviewers. As the deadline (Nov 26) is fast approaching, please let us know whether we have addressed all the issues.
> >
> > Thank you,

---

> > > ### Comment · Reviewer_AZT5 · 2024-11-25
> > >
> > > Thank you for your detailed responses, which have addressed most of my concerns. However, the method is highly sensitive to the pre-trained saliency and inpainting models, and the other two reviewers also raised concerns about this, resulting in negative ratings. While the end-to-end framework and the results are impressive, the work's limitations are quite clear. Consequently, I lowered the overall score to 5 (marginally below the acceptance threshold).

---

> ### Author Response · Authors · 2024-11-25
>
> Thank you for taking the time to review our work and for acknowledging the strengths of our end-to-end framework and results. We understand and respect your concern regarding the method's reliance on pre-trained saliency and inpainting models.
>
> We have responded to your concerns of these off-the-shelf models in W1/A1. To highlight it, we analyze the choice of these models and propose solutions to them (Figure 18 and Figure 19).
> - **If the inpainting model does not work well**. We show an example below where LAMA inpaints the non-saliency image with undesired textures. Luckily, as shown in L428-L431, Figure 6 and Figure 11, our model emerges with awareness of the affordance. It therefore places the content correctly and the undesired part is occluded. We include an example in **Figure 18** in our revised version.
> - **If the saliency detector fails:** for the saliency detector, we have demonstrated one limitation in Section 4.6. Also when there are no obvious salient objects, it may produce unreliable results. In that case, we can provide the model with an **all-one mask**, and our model becomes a cropping model. We show an example in **Figure 19** in our revised version.
>
> Also, we would like to emphasize that, saliency detectors are widely used in image retargeting even in recent works [2, 3, 4]. **We are not the only work that directly uses saliency maps.**
>
> [2] Shen, Feihong, et al. "Prune and Repaint: Content-Aware Image Retargeting for any Ratio." NeurIPS 2024.
>
> [3] Elsner, Tim, et al. "Retargeting Visual Data with Deformation Fields." ECCV 2024.
>
> [4] Valdez-Balderas, Daniel, Oleg Muraveynyk, and Timothy Smith. "Fast hybrid image retargeting." ICIP 2021.

---

> > ### Author Response · Authors · 2024-11-27
> >
> > Hi Reviewer AZT5,
> >
> > Thanks for your comments. Let us know whether we have addressed your concerns or not. Your reply will help AC to make a fair decision. Thanks!

---

### Official Review · Reviewer_itgh · 2024-11-03

**Soundness:** 3
**Presentation:** 2
**Contribution:** 3
**Rating:** 5
**Confidence:** 3

**Summary:**

The authors introduce  a novel image retargeting method that leverages a layer decomposition approach to distinguish between salient and non-salient regions of an image. By applying distinct warping fields to these separate layers, the method minimizes overall distortion during aspect ratio changes. Additionally, the authors present a novel structure-prone perceptual loss, based on DreamSim, which evaluates structural similarity between input and output images.

**Strengths:**

1) The authors address an interesting image rescaling problem and achieve visually impressive results compared to previous methods.

2) Their proposed multi-flow network for estimating transformations intuitively utilizes both the original undistorted image and a naively rescaled version to the target aspect ratio.

3) The proposed augmentation technique is logical and straightforward allowing to utilize DreamSim as loss function.

**Weaknesses:**

1) The overall contribution of the method appears somewhat limited, as the primary innovation lies in the augmentation technique that enables the use of the DreamSim loss. Meanwhile, the layer decomposition and inpainting modules are employed in a plug-and-play manner without addressing their sensitivity or taking further steps to reduce reliance on off-the-shelf methods.

2) The quantitative evaluation is somewhat limited; incorporating additional no-reference perceptual image metrics, both traditional and learning-based, would strengthen the paper's claims.

3) The writing quality and overall presentation should be enhanced to better reflect the promise of the proposed method's results.

4) The related work section requires improvement, particularly in highlighting the shortcomings of prior approaches and demonstrating how the proposed method overcomes the challenges outlined in the introduction.

**Questions:**

1) Why does GPNN outperform on the perceptual metric MUSIQ despite the provided visuals clearly showing that the proposed method delivers superior results?

2) How sensitive is the proposed approach to suboptimal inpainting results?

3) Why are the visual and quantitative results nearly identical when using either augmentation or the LPIPS loss? Does the model trained with the LPIPS loss not incorporate the augmentation method?

---

> ### Author Response · Authors · 2024-11-22
>
> Thank you for your valuable comments and suggestions. We have uploaded a **revised version** with updated results and comments. Please download the updated version on **the upper-right PDF button.**
>
> We address the comments below.
>
> **W1: Limited contributions**
> ---
> **A1**: We beg to differ. Our contribution is not limited to the augmentation technique for DreamSim. Our main contributions are as follows:
> - **To make our image retargeting network end-to-end trainable, we propose to use differentiable cross-attention mechanisms as replacement for PNNs**. Using PNN (Patch Nearest Neighbor) is one of the state-of-the-art methods (e.g., GPDM [3] and GPNN [4]) for image retargeting. PNN uses a non-differentiable NN manner to exchange the information between the image at target size and the image at original size, which is not learnable. To learn an end-to-end model, we replace the non-differentiable PNN with differentiable cross-attention blocks (Section 3.2). We are not aware of other retargeting methods incorporating PNN into their algorithms in a differentiable way.
> - **We propose to use layered transformations in our end-to-end model to preserve salient contents and avoid Out-of-Boundary (OOB) issues.** Previous retargeting methods either introduce OOB issues by using a single transformation (Figure 2, WSSDCNN [1]), or have difficulties preserving the content from the input (Figure 2, Shift-Map [2] and GPDM [3]).
> We carefully locate these limitations and propose a more flexible layered transformation way to mitigate the OOB issue.
> After the training, it gives the salient contents more significant transformation and non-salient contents minor transformation, which preserve the most important content.
> To the best of our knowledge, we are not aware of other retargeting methods using such a layered, end-to-end approach for image retargeting.
> - **We are the first work that makes DreamSim [5], a structure-sensitive loss, work for the image retargeting task**. We experiment with different perceptual loss functions (Figure 4 and Table 2), including LPIPS and DreamSim, and find DreamSim is more sensitive to the distortions than LPIPS. We further propose PSSL to adapt DreamSim for image retargeting.
> - **Our trained network enjoys fast inference speed**. Our HALO achieves a very fast inference speed after the training (Table 2). It is ~29,000x faster than sinDDM, an optimization-based method, and it is ~1.3x faster than WSSDCNN, an learning-based method.
>
> **W2: Limitations of the off-the-shelf methods.**
> ---
> **A2**: We acknowledge that these off-the-shelf methods, i.e., inpainting model and saliency detector, have their limitations.
>
> [**Inpainting model**]
>
> **Why LAMA?** For our inpainting model (LAMA), we use a dilated mask to improve the inpainting quality. We find it sufficient for our task. We here show a weaker inpaiting model, `cv2.inpaint`, to show LAMA’s superiority. We show the visualization in **Figure 17** in our revised version. LAMA shows better inpainting ability.
>
> **If LAMA does not work well.** We show an example below where LAMA inpaints the non-saliency image with undesired textures. Luckily, as shown in L428-L431, Figure 6 and Figure 11, our model emerges with awareness of the affordance. It therefore places the content correctly and the undesired part is occluded. We include an example in Figure 18 in our revised version.
>
> [**Saliency detector**]
>
> **Why MDSAM?** To demonstrate the effectiveness of the saliency detector, we also report the results where we do not use the saliency detector and let the model take an all-one mask as input and predict transformations. We report its performance below (also reported in **Table 5** in our revised version). This demonstrate that the effectiveness of MDSAM.
>
> | Model | CLIP sim.(%) | DreamSim sim.(%) | MUSIQ (%) | Average |
> |----------|----------|----------|----------|----------|
> |Single Transformation   | 88.33 | 80.8 | 47.9 |72.3
> |w/o saliency detector    | 87.25     | **82.1**     | 48.6     | 72.6     |
> | Ours (full)    | **90.17**   | 78.1     | **51.5**     | **73.3**     |
>
> **If MDSAM fails:** for the saliency detector, we have demonstrated one limitation in Section 4.6. Also when there are no obvious salient objects, it may produce unreliable results. In that case, we can provide the model with an **all-one mask**, and our model becomes a cropping model. We show an example in **Figure 19** in our revised version.
> We would like to emphasize that, for this challenging case (no obvious saliency), it is ill-posed and there are multiple solutions.

---

> ### Author Response · Authors · 2024-11-22
>
> [Cont.]
>
> **W3: More no-reference metrics.**
> ---
> **A3**: We additionally show a learning-based method, VILA [6], and a non-learning method, NIQE [7] below. Our method achieves competitive no-reference scores among three metrics and best avg. score. (**Table 4** in our revised paper). For NIQE, we normalize it to % and compute $100-norm(NIEQ)$.
>
> | Method                    | CLIP sim.(%) ↑ | DreamSim sim.(%) ↑ | MUSIQ(%) ↑ | VILA(%) ↑ | NIQE ↓ | Average(%) ↑ |
> |---------------------------|----------------|---------------------|------------|-----------|--------|--------------|
> | SINE  | 53.3           | 59.6               | 49.2       | 44.5      | 5.40   | 50.5         |
> | SinDDM  | 79.1       | 40.1               | 36.0       | 24.8      | 6.72   | 42.5         |
> | GPDM | 53.6         | 65.5               | 48.5       | 47.3      | 4.39   | 54.2         |
> | GPNN   | 88.5     | _77.5_             | 50.7       | **50.7**  | 4.74   | 64.0         |
> | Self-Play-RL | 88.7 | 76.2               | **52.1**   | **50.7**  | 4.40   | _64.8_       |
> | Cycle-IR   | 86.7         | 77.0               | 50.4       | 45.2      | 4.43   | 63.0         |
> | WSSDCNN    | 85.4         | 69.6               | 41.8       | 33.0      | 6.84   | 52.3         |
> | MagicFixup | 84.8      | 70.1               | 47.1       | 42.4      | 4.48   | 59.9         |
> | DragonDiffusion | _89.4_   | 66.8               | 51.1       | 47.1      | **3.96** | 62.9      |
> | **HALO (Ours)**       | **90.2**     | **78.0**           | _51.5_     | _48.1_    | _4.33_ | **64.9**     |
>
> **W4: promises of method**
> ---
> **A4:** Thank you for the suggestions. We highlight the promises of methods in A1 (our contributions).
>
> **W5: Related works**
> ---
> **A5:** Thank you for your suggestions. We would like to highlight our methods as follows in each part. We are open to suggestions. We also made the change in **Section 2** in the revised paper, labelled by red color.
>
> - **Image Retargeting.** Compared to optimization-based methods, we train an end-to-end model and it has faster inference speed. Compared to end-to-end methods, our method uses layered transformations and predicts multiple warping flows, avoiding out-of-boundary (OOB) issues.
> - **Layered representations.** We adopt the idea of layered representations and use it in the image retargeting task. It avoids out-of-boundary (OOB) issues in the previous methods.
> - **Perceptual losses.** We use DreamSim, a perceptual loss focusing on the mid-level features such as structures and layouts. We find previous perceptual loss functions (e.g., LPIPS) have difficulties handling structure distortions. We further adapt DreamSim to the image retargeting task by proposing an augmentation.
>
> **Q1: GPNN outperforms on the perceptual metric MUSIQ while it presents a poor visual result.**
> ---
> **A6:** We beg to differ. In Table 2, the MUSIQ score of **GPNN** is 50.7, which is **worse** than our method (51.5). **Self-Play-RL** does show a higher MUSIQ score of 52.1. However, we find MUSIQ itself may not be sensitive to the distortions as they are trained with undistorted images. We show an example in **Figure 20** in our revised version. Therefore, we include other metrics, e.g., DreamSim, CLIP and user study to further demonstrate the performance of each method.
>
> **Q2: Sensitivity of the inpainting model.**
> ---
> **A7:** We acknowledge that the inpainting model, LAMA, has its limitations. LAMA sometimes fails when the mask is large. We show an example in **Figure 18** in our revised paper, where LAMA inpaints the non-saliency image with undesired textures.  Luckily, as shown in L428-L431, Figure 6 and Figure 11, our model emerges with awareness of the affordance. It therefore places the content correctly and the undesired part is occluded.
>
> **Q3: LPIPS loss**
> ---
> **A8:** We do not use the augmentation in LPIPS, and this gives a similar result as “w/o augmentation” in Table 3.
> We include the result with LPIPS + augmentation below (also see **Table 6** in our revised paper).  With augmentation, LPIPS gets improved. However, its performance is still worse than DreamSim as LPIPS is not sensitive to the structure (Figure 4).
> We believe that our layout augmentation could serve as a “plug-in” module to improve the performance for the image retargeting task.
>
> | Method                          | CLIP sim.(%) ↑ | DreamSim sim.(%) ↑ | MUSIQ(%) ↑ |
> |---------------------------------|----------------|---------------------|------------|
> | Ours (w/ LPIPS)                 | 89.67          | 76.9               | 49.2       |
> | Ours (w/ LPIPS + aug.)          | _90.15_        | _77.2_             | _50.3_     |
> | Ours (w/ DreamSim)              | 89.69          | 76.9               | 48.9       |
> | Ours (w/ DreamSim + aug.)       | **90.17**      | **78.1**           | **51.5**   |

---

> > ### Author Response · Authors · 2024-11-22
> >
> > References
> > ---
> > [1] Cho, Donghyeon, et al. "Weakly-and self-supervised learning for content-aware deep image retargeting." ICCV 2017.
> >
> > [2] Pritch, Yael, Eitam Kav-Venaki, and Shmuel Peleg. "Shift-map image editing." ICCV 2009.
> >
> > [3] Elnekave, Ariel, and Yair Weiss. "Generating natural images with direct patch distributions matching." ECCV 2022.
> >
> > [4] Granot, Niv, et al. "Drop the gan: In defense of patches nearest neighbors as single image generative models." ECCV 2022.
> >
> > [5] Fu, Stephanie, et al. "Dreamsim: Learning new dimensions of human visual similarity using synthetic data." NeurIPS 2023.
> >
> > [6] Ke, Junjie, et al. "Vila: Learning image aesthetics from user comments with vision-language pretraining." Proceedings of the IEEE/CVF Conference on Computer Vision and Pattern Recognition. 2023.
> >
> > [7] Mittal, Anish, Rajiv Soundararajan, and Alan C. Bovik. "Making a “completely blind” image quality analyzer." IEEE Signal processing letters 20.3 (2012): 209-212.

---

> ### Author Response · Authors · 2024-11-24
>
> Hi Reviewer itgh,
>
> Thanks again for your valuable feedback to our paper. We have added more experimental results to address your comments. Let us know if you still have concerns or not. If you think your concerns are resolved. We kindly ask you to reply to this thread and change the score accordingly. If you still have concerns, let us know, we will add more results if needed. Your reply means a lot to us and helps to build a better research community. Thanks again.

---

> > ### Comment · Reviewer_itgh · 2024-11-24
> >
> > Dear Authors,
> >
> > Thank you for providing additional explanations and results in the rebuttal.
> >
> > However, my primary concern remains unchanged: the limited contribution beyond the clever engineering employed to build an end-to-end retargeting pipeline. As highlighted by the other reviewers, the dependence and sensitivity of your approach to the saliency detection and inpainting methods are significant limitations. As it stands, your work offers a promising baseline but does not fully address these critical dependencies.
> >
> > Have you considered integrating the saliency detection and/or inpainting processes into your framework to mitigate this issue?
> >
> > Regarding W1/A1, I believe that a discussion of the limitations of your contribution is still warranted. For instance, simply incorporating cross-attention or an alternative perceptual loss does not elevate the method beyond being a combination of prior works. While the visual results are impressive, the additional metrics fail to substantiate these outcomes. Why is there such a disparity?
> >
> > Overall, I view this paper as being halfway to a more complete work that adequately addresses the evident limitations stemming from its reliance on two external modules. This reliance was apparent from the outset and significantly impacts the originality of the contribution. Consequently, this form of incremental engineering feels insufficient for acceptance at this venue. Given that the other reviewers, especially 3SyR, share similar concerns and the new results provided do not substantially strengthen the paper, I am inclined to recommend rejection.

---

> ### Author Response · Authors · 2024-11-25
>
> We sincerely appreciate your thoughtful feedback and your assessment of our work. Below, we address your concerns in detail.
>
> > Saliency maps & inpainting
> - Image retargeting is a challenging task and saliency maps are widely recognized as crucial for it, as the saliency information directly informs the preservation of the critical content.
> - Two recent works, one from NeurIPS 2024 [8] and the other from ECCV 2024 [9] (code released or published after the ICLR submission deadline) , directly use pretrained saliency detectors. **We believe saliency maps are widely adopted in image retargeting research and we are not the only case that uses saliency maps.**
> - Furthermore,  we have already acknowledged and discussed one of the limitations of the saliency detectors in the original manuscripts. Improving saliency detectors is a parallel work to ours.
> - We show that without the saliency detectors, the performance drops. Also many other works (Table 2 & 3, including generative models SINE, SinDDM, MagicFixup and DragonDiffusion) without saliency maps cannot achieve the same performance as ours.
> | Method            | Accuracy (%) |
> |-------------------|--------------|
> | SINE             | 54.0         |
> | SinDDM           | 51.7         |
> | MagixFixup       | 67.3         |
> | DragonDiffusion  | 69.1         |
> | Ours             | **73.2**         |
>
> User study:
> | Method            | User preference (%) |
> |-------------------|--------------|
> | GPDM             | 5.47         |
> | MagixFixup       | 30.23         |
> | Ours       | **43.44**       |
>
> - We carefully propose our solution to saliency detector’s failure cases (Figure 19) in the rebuttal.
>
>
> > Metrics
> - To the best of our knowledge, there are **no good metrics for image retargeting research**. People still heavily **rely on user studies** to show algorithm improvement.
> -  We conducted **a comprehensive user study**,  which we believe is the gold standard for evaluating the perceptual quality of retargeting results. Our user study shows that our method outperformed other methods by **a large margin of 13.21%**.
> - The **disparity** between objective metrics and visual outcomes is **a well-known challenge in vision tasks**. For example, metrics like PSNR and SSIM often prioritize pixel-wise accuracy, which can inadvertently favor blurry results, as highlighted in related works [10, 11, 12]. **Rather than focusing solely on these metrics, we prioritize perceptual quality and user studies as more reliable indicators of success in image retargeting**. We believe that the development of more suitable metrics for this task remains an open area for discussion and future exploration.
>
> > Contributions/novelty
> - Image retargeting is a **highly challenging task**, full of practical applications across industries. The complexity lies in maintaining both semantic and structural fidelity under diverse transformations, which our method successfully tackles.
> - **Engineering work. We disagree with this.** We carefully analyze existing limitations (e.g., Figure 2) and systematically address them through proposed improvements by using proposed PSSL and multi-flow network. Rather than randomly integrating components, we designed and evaluated solutions that are informed by these analyses to ensure meaningful advancements. For example, directly using DreamSim does not work. We then propose the PSSL loss with the layout disturbance to make it work. Similar to the PNN’s non-differentiable blocks, we use a cross-attention to make it differentiable. We believe these proposed modules can also be utilized in other works not only in image retargeting.
>
> > References
>
> [8] Shen, Feihong, et al. "Prune and Repaint: Content-Aware Image Retargeting for any Ratio." NeurIPS 2024.
>
> [9] Elsner, Tim, et al. "Retargeting Visual Data with Deformation Fields." ECCV 2024.
>
> [10] Rota, Claudio, Marco Buzzelli, and Joost van de Weijer. "Enhancing Perceptual Quality in Video Super-Resolution through Temporally-Consistent Detail Synthesis using Diffusion Models." ECCV 2024.
>
> [11] Kang, Minguk, et al. "Scaling up gans for text-to-image synthesis." CVPR 2023.
>
> [12] Rombach, Robin, et al. "High-resolution image synthesis with latent diffusion models." CVPR 2022.

---

> > ### Author Response · Authors · 2024-11-27
> >
> > Hi Reviewer itgh,
> >
> > Thanks for your comments. Let us know whether we have addressed your concerns or not. Your reply will help AC to make a fair decision. Thanks!

---

### Official Review · Reviewer_3SyR · 2024-11-07

**Soundness:** 2
**Presentation:** 2
**Contribution:** 2
**Rating:** 3
**Confidence:** 5

**Summary:**

The work uses saliency to achieve image retargeting in a learnable setting. The results are shown to be SOTA through an user study.

**Strengths:**

Nil

**Weaknesses:**

- The work relies heavily on a saliency detector - MDSAM, which is a challenge. Having the main task of retargeting rely on a pre-trained saliency detection network is a downer. The work could have been better with solid contributions than just assembling pre-trained models for other tasks.

- RetargetMe dataset is a standard one. However, the images are very simple and have distinct objects. The proposed work will fail when there are complex scenes with nonsalient objects.

- Using saliency for retargeting has been explored extensively since the work of seam carving. There is hardly any novelty or contribution in this work.

**Questions:**

See above.

---

> ### Author Response · Authors · 2024-11-22
>
> Thank you for your comments.  We have uploaded a **revised version** with updated results and comments. Please download the updated version on **the upper-right PDF button.**
>
> We address your concerns below.
>
> **W1: Limitations of the saliency detector.**
> ---
> **A1:** We see that saliency maps have been widely used in recent image retargeting works  [1, 2, 3].
>
> However, we acknowledge that the saliency detector has its limitations. We address these concerns below.
>
> **Why MDSAM?** To demonstrate the effectiveness of the saliency detector, we also report the results where we do not use the saliency detector and let the model take an all-one mask as input and predict transformations. We report its performance below (also reported in **Table 5** in our revised version). This demonstrate that the effectiveness of MDSAM.
>
> | Model | CLIP sim.(%) | DreamSim sim.(%) | MUSIQ (%) | Average |
> |----------|----------|----------|----------|----------|
> |Single Transformation   | 88.33 | 80.8 | 47.9 |72.3
> |w/o saliency detector    | 87.25     | **82.1**     | 48.6     | 72.6     |
> | Ours (full)    | **90.17**   | 78.1     | **51.5**     | **73.3**     |
>
> **If MDSAM fails:** for the saliency detector, we have demonstrated one limitation in Section 4.6 in our paper. Also when there are no obvious salient objects, it may produce unreliable results. In that case, we can provide the model with an **all-one mask**, and our model becomes a cropping model. We show an example in **Figure 19** in our revised version.
> We would like to emphasize that, for this challenging case (no obvious saliency), it is ill-posed and there are multiple solutions.
>
> **W2: RetargetMe is very simple.**
> ---
> **A2:** We respectfully disagree on this. RetargetMe is the most common evaluation benchmark for evaluating retargeting algorithms. It has diverse data categories including animals, buildings, landscapes and humans. We are not aware of other benchmarks for evaluating the retargeting task. One mostly recent work [1] (will be published at NeurIPS 2024 after the ICLR submission deadline) also evaluates the approach on RetargetMe. It is not necessarily a weakness to consider evaluation on the most common dataset.
> In addition to RetargetMe, we show extensive results on the in-the-wild data in Figure 8, Figure 14, Figure 15 and FIgure 16, demonstrating the effectiveness of our method even to unseen data.
>
> **W3: Using saliency is not novel.**
> ---
> **A3:** **We do not claim using saliency as one of contributions.** The contributions of our work include:
> - **To make our image retargeting network end-to-end trainable, we propose to use differentiable cross-attention mechanisms as replacement for PNNs.** Using PNN (Patch Nearest Neighbor) is one of the state-of-the-art methods (e.g., GPDM [6] and GPNN [7]) for image retargeting. PNN uses a non-differentiable NN manner to exchange the information between the image at target size and the image at original size, which is not learnable. To learn an end-to-end model, we replace the non-differentiable PNN with differentiable cross-attention blocks (Section 3.2). We are not aware of other retargeting methods incorporating PNN into their algorithms in a differentiable way.
> - **We propose to use layered transformations in our end-to-end model to preserve salient contents and avoid Out-of-Boundary (OOB) issues.** Previous retargeting methods either introduce OOB issues by using a single transformation (Figure 2, WSSDCNN [4]), or have difficulties preserving the content from the input (Figure 2, Shift-Map [5] and GPDM [6]).
> We carefully locate these limitations and propose a more flexible layered transformation way to mitigate the OOB issue.
> After the training, it gives the salient contents more significant transformation and non-salient contents minor transformation, which preserve the most important content.
> To the best of our knowledge, **we are not aware of other retargeting methods using such a layered, end-to-end approach for image retargeting.**
> - **We are the first work that makes DreamSim [8], a structure-sensitive loss, work for the image retargeting task.** We experiment with different perceptual loss functions (Figure 4 and Table 2), including LPIPS and DreamSim, and find DreamSim is more sensitive to the distortions than LPIPS. We further propose PSSL to adapt DreamSim for image retargeting.
> - **Our trained network enjoys fast inference speed. Our HALO achieves a very fast inference speed after the training (Table 2).** It is ~29,000x faster than sinDDM, an optimization-based method, and it is ~1.3x faster than WSSDCNN, an learning-based method.

---

> ### Author Response · Authors · 2024-11-22
>
> References
> ---
> [1] Shen, Feihong, et al. "Prune and Repaint: Content-Aware Image Retargeting for any Ratio." NeurIPS 2024.
>
> [2] Elsner, Tim, et al. "Retargeting Visual Data with Deformation Fields." ECCV 2024.
>
> [3] Valdez-Balderas, Daniel, Oleg Muraveynyk, and Timothy Smith. "Fast hybrid image retargeting." ICIP 2021.
>
> [4] Cho, Donghyeon, et al. "Weakly-and self-supervised learning for content-aware deep image retargeting." ICCV 2017.
>
> [5] Pritch, Yael, Eitam Kav-Venaki, and Shmuel Peleg. "Shift-map image editing." ICCV 2009.
>
> [6] Elnekave, Ariel, and Yair Weiss. "Generating natural images with direct patch distributions matching." ECCV 2022.
>
> [7] Granot, Niv, et al. "Drop the gan: In defense of patches nearest neighbors as single image generative models." ECCV 2022.
>
> [8] Fu, Stephanie, et al. "Dreamsim: Learning new dimensions of human visual similarity using synthetic data." NeurIPS 2023.

---

> > ### Comment · Reviewer_3SyR · 2024-11-23
> > **Increased the rating a bit**
> >
> > The rebuttal regarding the dataset and saliency does not convince me. Reliance on one saliency detector on a standard dataset adds no value. Given the authors' efforts to clarify my concerns, I increased the rating a bit, but still, I am negative.

---

> ### Author Response · Authors · 2024-11-24
>
> Thanks for your considerations to raise the rating. We would like to have more discussions about your concerns on the saliency detector and the dataset.
>
> **Saliency detector**
> - We include the analysis of our saliency detector, MDSAM, in A1. We analyze **its importance** by removing it from our pipeline. We find without the saliency map, the performance drops (73.3->72.6), indicating that having a salient detector is helpful.
> - Furthermore, in A1, we propose **a solution to the failure cases** of it – by feeding the model **an all-one mask** – our model becomes a cropping model and still works for the case where saliency map is not obvious (**Figure 19**). This mitigates the full reliance on the saliency detector. Also, we believe this demonstrates that our model **is capable of different choices of saliency maps.**
> - We would like to emphasize the definition of image retargeting, that is to preserve the salient content and structure of the input image at the new resolution, with a minimum of visual artifacts. **Using saliency map is an intuitive and straightforward way to understand the salient content of the input image.** Also, recent papers [1,2,3] all use the saliency maps. We respectfully disagree with the idea that using saliency detectors is problematic.
> - We would like to emphasize again that we do **not** claim using saliency detectors as our contribution. Our paper addresses the poor preservation ability from previous retargeting methods by using layered transformations and layout augmentation for PSSL, which significantly improve the content and structure preservation (Table 3) and outperforming other state-of-the-art methods (Table 2)
> - Finally, for the cases where there is **no salient content**, it is an extremely ill-posed problem and there could be multiple solutions. We would like to emphasize that "no obvious saliency" is equivalent to "everything is salient". The problem becomes ill-posed because many solutions are reasonable.

---

> ### Author Response · Authors · 2024-11-24
>
> **Dataset**
> - We test our method on the standard benchmark, RetargetMe, as many previous works [1,2,3]. **We are not aware of other common datasets for image retargeting.**
> - To demonstrate our method generalizes well to other datasets, we have included qualitative results on the in-the-wild data from Unsplash dataset in **Figure 8, Figure 14, Figure 15 and Figure 16**. The Unsplash dataset consists of much diverse data and complicated scenarios. Our method, without finetuning, can already work well on many complex images.
> - We are now running a quantitative evaluation on the Unsplash dataset. We will post the results soon by tomorrow.
>
> References
>
> [1] Shen, Feihong, et al. "Prune and Repaint: Content-Aware Image Retargeting for any Ratio." NeurIPS 2024.
>
> [2] Elsner, Tim, et al. "Retargeting Visual Data with Deformation Fields." ECCV 2024.
>
> [3] Valdez-Balderas, Daniel, Oleg Muraveynyk, and Timothy Smith. "Fast hybrid image retargeting." ICIP 2021.

---

> ### Author Response · Authors · 2024-11-24
>
> [**Dataset cont.**]
>
> We present updated quantitative results on the Unsplash dataset below. This dataset comprises images sourced from the Internet. It consists of **diverse image categories**, including **complicated scenarios**.
>
> We randomly select 200 images (2.5$\times$ bigger than RetargetMe) from the dataset to evaluate various methods. **All of these data are unseen during training.**
>
> **Our method achieves the highest average score across three metrics**, outperforming other methods in all metrics except for the MUSIQ score, and this is achieved **without any fine-tuning**. Meanwhile, please also refer to **Figure 8, Figure 14, Figure 15 and Figure 16** for qualitative results.
>
> | Model           | MUSIQ | CLIP Sim. | DreamSim Sim. | Average |
> |------------------|-------|-----------|---------------|---------|
> | GPDM            | 43.26 | 53.20     | 52.64         | 49.70   |
> | Self-Play-RL    | **45.58** | 90.50     | 72.21         | 69.43   |
> | MagicFixup      | 44.70 | 89.94     | 65.48         | 66.71   |
> | Ours            | 45.42 | **91.18**     | **76.73**         | **71.11**   |
>
> We hope this helps you to resolve your concerns about the dataset and the saliency detector.

---

> > ### Author Response · Authors · 2024-11-25
> >
> > Hi Reviewer 3SyR,
> >
> > Could you please check our reply? Let us know if you still have concerns or not? Your reply means a lot to us. Thanks!

---

> > > ### Author Response · Authors · 2024-11-27
> > >
> > > Hi Reviewer 3SyR,
> > >
> > > Do you have additional comments? Your reply will help AC to make a fair decision. Thanks!

---

### Author Response · Authors · 2024-11-23

We highly appreciate the suggestions and comments from all the reviewers. Especially, our paper
- shows impressive results on image retargeting method (Reviewers itgh, AZT5)
- uses intuitive design for multi-flow network (Reviewer itgh)
- is logically sound to use augmentation (Reviewer itgh)
- is well-written with a clear and coherent structure (Reviewer AZT5).

We have uploaded a **revised version** with updated results and comments. Please download the updated version on **the upper-right PDF button.**
We now address the concerns from reviewers one by one.

---

### Meta-Review · Area_Chair_noDu · 2024-12-16

**Metareview:**

The paper received mixed reviews from three experts.

The authors provided responses and tried to persuade the reviewers to elevate their ratings.

The reviewers were not convinced by the authors' responses and reached consensus (reviewer AZT5 lowered to 5) on the negative side and recommend rejection and AC agrees with them.

The authors are invited to benefit from the received feedback and to further improve their work.

**Additional Comments On Reviewer Discussion:**

The authors tried to provide answers and new experimental results to address the concerns raised by the reviewers.

Reviewer 3SyR appreciates part of the provided new information and slightly improves the rating while still remaining negative, as "The rebuttal regarding the dataset and saliency does not convince me. Reliance on one saliency detector on a standard dataset adds no value."

Reviewer itgh points out the limited contributions, also after reading the authors' responses: "However, my primary concern remains unchanged: the limited contribution beyond the clever engineering employed to build an end-to-end retargeting pipeline. As highlighted by the other reviewers, the dependence and sensitivity of your approach to the saliency detection and inpainting methods are significant limitations. As it stands, your work offers a promising baseline but does not fully address these critical dependencies."

Reviewer AZT5 while satisfied by most of the responses, agrees with the other reviewers and lowers his initial rating
"However, the method is highly sensitive to the pre-trained saliency and inpainting models, and the other two reviewers also raised concerns about this, resulting in negative ratings. While the end-to-end framework and the results are impressive, the work's limitations are quite clear."

---

### Decision · Program_Chairs · 2025-01-22

Reject